nature
structural &
molecular biology
# Identifying amyloid-related diseases by mapping mutations in low-complexity protein domains to pathologies

Kevin A. Murray[1], Michael P. Hughes[1], Carolyn J. Hu[1], Michael R. Sawaya [1], Lukasz Salwinski [1], Hope Pan[1], Samuel W. French[2], Paul M. Seidler[3] and David S. Eisenberg [1]✉

Proteins including FUS, hnRNPA2, and TDP-43 reversibly aggregate into amyloid-like fibrils through interactions of their low-complexity domains (LCDs). Mutations in LCDs can promote irreversible amyloid aggregation and disease. We introduce a computational approach to identify mutations in LCDs of disease-associated proteins predicted to increase propensity for amyloid aggregation. We identify several disease-related mutations in the intermediate filament protein keratin-8 (KRT8). Atomic structures of wild-type and mutant KRT8 segments confirm the transition to a pleated strand capable of amyloid formation. Biochemical analysis reveals KRT8 forms amyloid aggregates, and the identified mutations promote aggregation. Aggregated KRT8 is found in Mallory–Denk bodies, observed in hepatocytes of livers with alcoholic steatohepatitis (ASH). We demonstrate that ethanol promotes KRT8 aggregation, and KRT8 amyloids co-crystallize with alcohol. Lastly, KRT8 aggregation can be seeded by liver extract from people with ASH, consistent with the amyloid nature of KRT8 aggregates and the classification of ASH as an amyloid-related condition.

Protein aggregation is a pathological characteristic of dozens of diseases, including Alzheimer's disease (AD), Parkinson's disease (PD), amyotrophic lateral sclerosis (ALS), prion diseases, systemic amyloidosis, and even certain metabolic diseases, such as type II diabetes. In these disease contexts, normally soluble, globular proteins assemble into elongated insoluble fibrils, known as amyloid fibrils[1]. Amyloid fibrils deposit within tissues, either intra- or extracellularly, where they interfere with native cellular functions, elicit toxicity, and contribute to pathogenesis. Atomic structures of amyloid fibrils reveal stacked, identical layers of protein monomers that form beta-sheets. Segments of the beta-sheets of the fibril core tightly interdigitate, forming structural motifs known as steric zippers, in which the amino acid side chains interlock like the teeth of a zipper down the fibril axis[2]. These highly stable interactions likely contribute to the resistance of amyloid fibrils to disaggregation and proteolytic degradation. Analysis of sequences capable of adopting steric-zipper conformations has successfully identified protein segments that drive amyloid formation[3].

Whereas protein aggregation was once believed to be strictly pathogenic, growing evidence suggests that proteins also undergo a functional amyloid-like form of aggregation to drive cellular processes. These states of protein self-assembly are often reversible and contribute to the formation of dynamic membraneless cellular bodies, such as stress granules, P-bodies, Cajal bodies, and nuclear paraspeckles[4,5]. Unlike disease-associated amyloid aggregates, these functional assemblies associate and dissociate reversibly. This reversibility may be driven by specific structural motifs in protein low-complexity domains (LCDs). Termed LARKS (low-complexity amyloid-like kinked segments), these motifs form fibrillar structures with a kinked beta-strand conformation as opposed to the predominantly pleated strands of steric zippers in

pathogenic amyloid fibrils[6]. In both reversible and pathogenic amyloid, identical beta-strands of different protein monomers, kinked or pleated, are stacked to form beta-sheets that run the length of the fibril. The proteins FUS, TDP-43, and hnRNPA2 are enriched in LARKS and are all known to reversibly phase separate[7–9], and cryo-EM/ssNMR fibril structures of FUS and hnRNPA2 exhibit a high degree of kinking within the beta-sheets[10,11]. However, each of these proteins is known to aggregate irreversibly in several diseases, and studies have shown that fibrous FUS aggregates can arise from phase-separated droplets[12,13].

The molecular mechanisms that drive the transition from reversible to irreversible amyloid aggregation are incompletely understood. A number of pathogenic mutations within the LCDs of proteins known to undergo both reversible and irreversible self-assembly have been demonstrated to increase aggregation propensity, including in FUS[14], hnRNPA1 and hnRNPA2 (ref. [12]), and TDP-43 (ref. [15]). Here we seek to better understand this transition from reversibility to irreversibility through structural and biochemical characterization of disease-related mutations occurring within LARKS. We have developed a computational screen to identify mutations in LARKS that may drive the transition from a kinked beta-sheet conformation to a pleated one. Our computational approach identifies many known aggregation-promoting mutants while finding new variants not previously associated with protein aggregation.

Among these new variants are mutations in the low-complexity region of the intermediate filament protein KRT8. Recent reports have demonstrated that low-complexity domains in several intermediate filament proteins undergo amyloid assembly[16,17]. We demonstrate that specific LARKS mutations in KRT8 promote both its reversible and irreversible aggregation, highlighting the amyloid-like nature of KRT8 aggregation. Atomic structures of KRT8 LARKS

[1]Departments of Chemistry and Biochemistry and Biological Chemistry, UCLA-DOE Institute, Molecular Biology Institute, and Howard Hughes Medical Institute, UCLA, Los Angeles, CA, USA. [2]Department of Pathology & Laboratory Medicine, David Geffen School of Medicine at UCLA, Los Angeles, CA, USA. [3]Department of Pharmacology and Pharmaceutical Science, University of Southern California, Los Angeles, CA, USA. ✉e-mail: david@mbi.ucla.edu

reveal that pathogenic mutations convert the non-pleated wild-type protein into a pleated conformation, shedding light on the molecular mechanisms underlying aggregation-promoting mutations in low-complexity proteins. Furthermore, aggregated KRT8 is present in Mallory–Denk bodies (MDBs), found in a variety of liver diseases but most predominantly in ASH. We show that the amyloid aggregation of KRT8 can be seeded by ASH liver tissue extracts, implicating amyloid interactions in alcoholic liver disease.

## Results

**Computational screen of disease-related mutations in LARKS.** Irreversibly formed amyloid fibrils are generally characterized by pleated beta-sheets, and reversible LARKS by kinked beta-sheets. Thus, we hypothesize that mutations in reversibly aggregating proteins may promote pathogenic irreversible aggregation by driving a transition from a kinked to pleated beta-sheet conformation, favoring formation of steric zippers. To study this, we first applied computational threading to identify LARKS, the kinked segments thought to drive reversible self-assembly[18]. Computational threading determines whether a given amino acid sequence is compatible with a particular atomic structure, in this case LARKS. We fit (or 'threaded') the sequences of the human proteome onto the kinked backbones of three LARKS structures (Fig. 1a) and used the protein-modeling software Rosetta to assess the energy of each structure. A sequence is accepted as one that forms LARKS if it fits one of the three LARKS structures with a sufficiently low energy. This threading approach has been successfully used to identify steric-zipper segments in amyloid proteins[3].

Next, to identify disease-related mutations within the LARKS, we cross-referenced the LARKS with three mutational databases: OMIM[19], Uniprot[20], and ClinVar[21]. As a control, benign single-nucleotide polymorphisms (SNPs) occurring within the LARKS sequences were also analyzed. Analysis of the mutations reveals pathogenic mutations are mainly from glycine or proline to either polar or hydrophobic residues (Fig. 1c), whereas benign SNPs lead to less disruptive changes in amino acid properties (for example, Gly to Ser) (Extended Data Fig. 1).

We add a final computational step to select the most likely mutations within LARKS that lead to disease. In this step, we assess whether a mutation increases the propensity of the sequence to form a pleated sheet as opposed to a kinked one. We thread the wild-type and mutant LARKS sequences onto the backbone of a pleated steric zipper. The calculated energies of the threaded zipper structures versus the difference in energy between the wild-type and mutant zippers are plotted (Fig. 1b). From this, we observe a population of LARKS mutations that shows a decrease in zipper energy (that is, more likely to form a zipper) in both their absolute score as well as compared with the wild type. This cluster is also relatively free of benign SNPs. Within this population, we find many mutations that have been previously identified to increase aggregation propensity (Fig. 1d). These mutations include TDP-43-G294V[15], TDP-43-G294A[9,15], TDP-43-G295S[9,15], and hnRNPA2-D290V[12,22] (Table 1). We also observe a large enrichment of mutations occurring in the head domain of the intermediate filament protein KRT8.

Although other intermediate filament proteins have been demonstrated to form amyloids[16,17,23], KRT8 specifically has not previously been characterized as an amyloid-forming protein. This protein, like other alpha-keratin proteins, contains a central coiled-coil domain that is flanked by an amino-terminal head domain and a carboxy-terminal tail domain, which are both low complexity in sequence composition[24]. The KRT8 mutants convert glycines to alanine, cysteine, or valine, and tyrosines to either histidine or cysteine. Both glycine and tyrosine have been identified as critical residues for maintaining liquid–liquid phase separations of low-complexity proteins[25]. For experimental validation, we selected three of these KRT8 mutants (G62C, Y54H, and G55A), all of which

are found in people with cryptogenic liver disease[26], as well as three FUS mutants (G191S, G225V, and G230C).

**Structures of wild-type and mutant LARKS of KRT8.** To assess the structural impacts of the disease-associated mutations, we determined the atomic structures of several LARKS from KRT8 for both the wild-type and mutant sequences. Segment crystal structures of KRT8$_{58-64}$ (wild type: SGMGGIT; G62C: SGMGCIT) for the wild-type protein and KRT8-G62C were determined. Similarly, the structure of KRT8$_{52-58}$ (GGYAGAS) with the G55A mutation was also determined. High-resolution diffraction data were collected for the wild-type sequence (GGYGGAS) of KRT8$_{52-58}$, but we were unable to determine the structure, presumably owing to its adoption of a highly kinked conformation. Fortunately, a recently determined structure from hnRNPA1 (GGYGGS) very closely resembles the wild-type KRT8$_{52-58}$ sequence, particularly the residues at and surrounding the mutation site, providing the needed comparison[27].

Both pairs of structures suggest that the mutant sequences form stronger interactions. Wild-type KRT8$_{58-64}$ (SGMGGIT) crystallized as an anti-parallel class-7 steric zipper, with a long PEG 1000 molecule binding along the fibril axis in the space between beta-sheets (Fig. 2a)[2]. Two ethanol molecules are observed, coordinated near the carboxy terminus. The interface appears to be formed by the hydrophobic interactions of the methionine and isoleucine with the tyrosine and backbone of the adjacent sheet. At glycine 62 (and glycine 61), we observe a highly extended backbone conformation, in which the dihedral angles at both residues approach nearly |180°|. In contrast, the mutated structure of KRT8$_{58-64}$ G62C (SGMGCIT) adopts a nearly completely pleated beta-sheet conformation (Fig. 2b). Forming a class-2 parallel steric zipper, the sheets associate again through hydrophobic interactions with the methionine and isoleucine but are further stabilized by interaction with the mutant cysteine. Isopropanol is also observed coordinated in the zipper interface. Importantly, instead of the extended conformation adopted by the glycine in the wild-type structure, the mutant cysteine is in a pleated conformation typical of most irreversible amyloid proteins. The wild-type structure is composed of extended, nearly linear beta-strands that result in a flattened beta-sheet. We hypothesize that this flattened conformation may lead to a less stable fibril structure, resulting from reduced side-chain interdigitation seen in typical steric zippers, which lowers the total buried surface area between interfacing strands. It is possible that this flattened conformation destabilizes the hydrogen bonding that occurs between layers of the fibril, and it has previously been demonstrated that an extended beta-strand geometry produces weaker inter-strand hydrogen bonding[28]. However, the mutant structure lacks this potentially destabilizing extended conformation, with extensive interdigitation between side chains within the steric-zipper interface. In short, although both wild-type and mutant structures are steric-zipper-like, the mutated structure appears to be more stabilizing.

The KRT8$_{52-58}$-G55A (GGYAGAS) structure forms an anti-parallel class-6 steric zipper, with sheets interacting through the tyrosine and two alanine residues, including the G55A mutant alanine (Fig. 2c). The glycine at the N terminus assumes a highly extended conformation, like the SGMGGIT structure. But, like the other mutant structure, the backbone at the mutant alanine is pleated. Comparison with GGGYGGS from hnRNPA1, which has the same GGYGG sequence motif as wild-type KRT8$_{52-58}$ (GGYGGAS), reveals that a glycine at that position adopts a highly extended backbone conformation. An overlay of both wild-type–mutant structure pairs highlights that mutations from wild-type glycines facilitate a transition in the backbone from a non-ideal, highly extended conformation to a pleated one (Fig. 2d,e). From these fibril structures, we observe that both the G62C and G55A mutations in KRT8 convert a non-pleated beta-sheet into a pleated one, transitioning a LARKS-like conformation into a steric zipper.

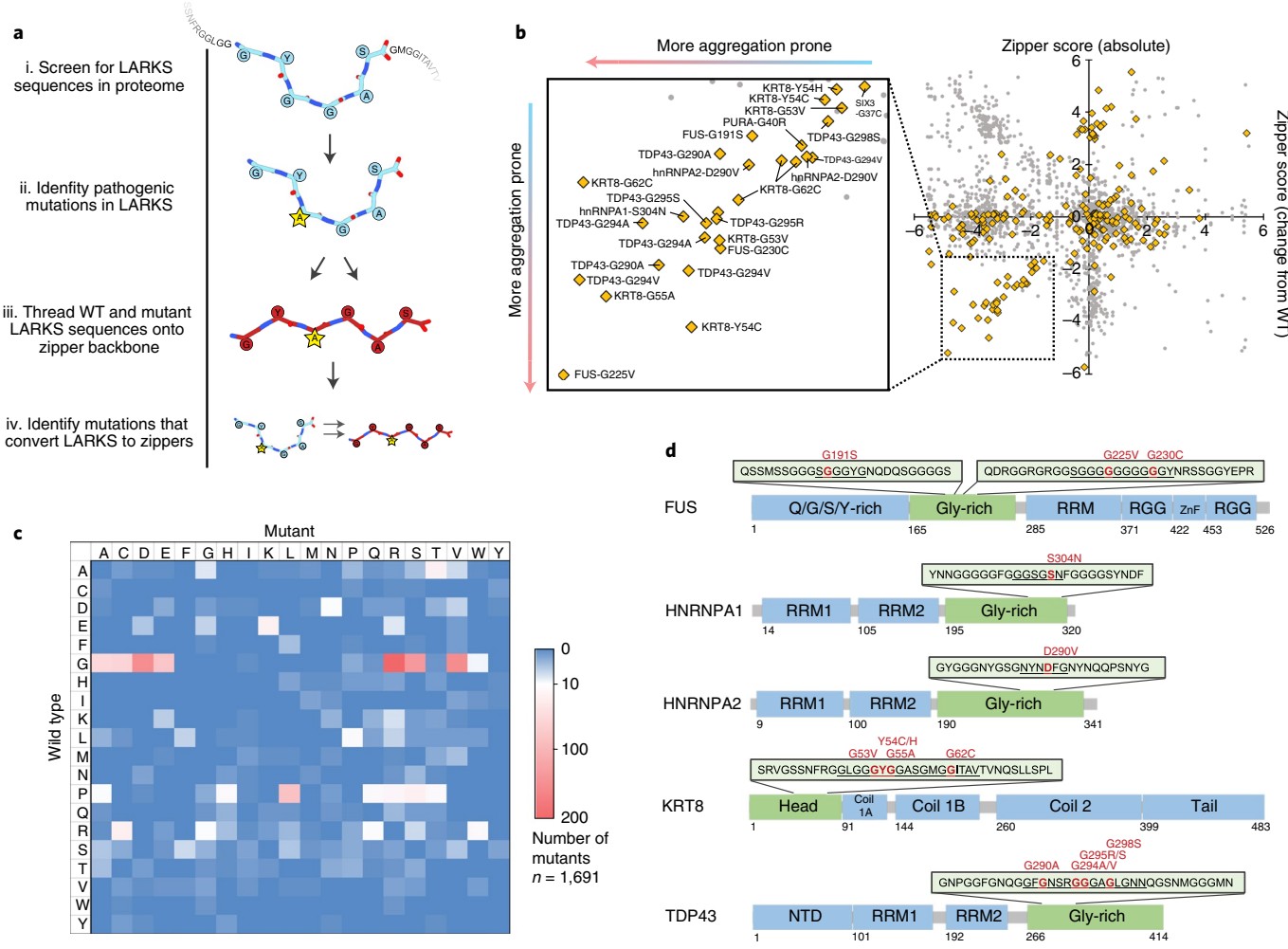

**Fig. 1 | Computational search for pathogenic mutations that potentiate amyloid aggregation. a**, Method for screening for mutations in LARKS that enable the conversion to a steric zipper. (i) The human proteome was screened for sequences capable of adopting LARKS-like structures, a structural motif underlying reversible aggregation. (ii) Mutational databases were queried to identify disease-linked variants that occur in the identified LARKS regions. (iii) The wild-type and mutant LARKS sequences were scored for their capacity to form steric zippers, the core structural motif underlying irreversible amyloid aggregation. (iv) Mutations in LARKS sequences that increase their propensity to form steric zippers were identified and further characterized. **b**, Plots of the steric-zipper scores for pathogenic mutations in LARKS (yellow) compared with benign SNPs (gray). The insert highlights a population of mutations with an increased likelihood of zipper formation. Many known mutations linked to amyloid aggregation were identified, including those in FUS, TDP-43, hnRNPA1, and hnRNPA2. **c**. Heat map of pathogenic mutations occurring in LARKS (n=1,691). Transition of wild-type glycine to polar or hydrophobic residues, and proline to leucine mutations, are the most commonly observed variants. **d**, Selected proteins with LARKS containing pathogenic mutations predicted to increase zipper formation, highlighted in red. Like proteins already characterized to reversibly and irreversibly aggregate (FUS, TDP-43, and hnRNPA), the computational screen also identified zipper-promoting mutations in LARKS in the intermediate filament protein KRT8, suggesting that it is in the same class as the other proteins.

**Characterization of disease-associated KRT8 mutations.** Our structural analysis highlighted a key difference in the backbone conformations between the wild-type and mutant KRT8 fibrils. We next studied the effects that these mutations have on KRT8 aggregation, finding that the wild type is slightly less prone to aggregation than are the mutants. Analysis of the KRT8 sequence using the protein disorder prediction server (PrDOS), which predicts natively disordered protein regions, identifies the head (residues 1–90) and tail (residues 399–483) domains as likely disordered (Fig. 3a). We expressed and purified the head domain of KRT8 (residues 1–90) with an N-terminal mCherry solubility tag for the wild-type protein and the G62C, Y54H, and G55A mutants. Aggregation kinetics assays using the fluorescent amyloid dye thioflavin T (ThT) reveal that extended shaking at 37 °C results in the aggregation of all four KRT8$_{1-90}$ constructs; however, the three mutants aggregated much more quickly and extensively than did the wild type (Fig. 3b and

Extended Data Fig. 2). Electron micrographs of each sample reveal clumped aggregates with similar morphologies for both the wild type and the mutants (Fig. 3c). X-ray powder diffraction of each sample displays a cross-beta diffraction pattern that is typical of amyloid fibrils. For all four samples, we observe diffraction rings near 10 Å and ranging between 4.3 Å and 4.6 Å, typical of the diffraction of LARKS and low-complexity protein aggregates[6,25] (Fig. 3d and Extended Data Fig. 3). We analyzed stability of the aggregates, finding that the mutants are slightly more resistant to SDS denaturation than is the wild-type protein (Extended Data Fig. 4).

Having observed the effects of each mutation on irreversible aggregation, we next asked whether these mutations promote reversible keratin aggregation, finding that they do. Keratins and other intermediate filament proteins have been shown to phase separate reversibly both in vitro and in cells[23]. To quantitatively measure the effects of each mutant on phase separation, we developed

**Table 1 | List of pathogenic mutations in LARKS identified to increase the likelihood of steric-zipper formation**

| protein | Mutation residue no. | Wild type | Mutant | Predicted LARKS | Predicted zipper |
|---|---|---|---|---|---|
| FUS | 191 | G | S | SGGGYG | SSGGYG |
| | 225 | G | V | SGGGGG | SGGGVG |
| | 230 | G | C | GGGGGY | GGGCGY |
| GATAD1 | 59 | G | A | GFGAAT | AFGAAT |
| hnRNPA1 | 304 | S | N | GGSGSN | GGSGNN |
| hnRNPA2 | 290 | D | V | SGNYND | SGNYNV |
| | 290 | D | V | NYNDFG | NYNVFG |
| KRT8 | 53 | G | V | GLGGGY | GLGGVY |
| | 53 | G | V | LGGGYG | LGGVYG |
| | 54 | Y | C | YGGASG | CGGASG |
| | 54 | Y | C | GYGGAS | GCGGAS |
| | 54 | Y | H | GYGGAS | GHGGAS |
| | 55 | G | A | GYGGAS | GYAGAS |
| | 62 | G | C | MGGITA | MGCITA |
| | 62 | G | C | SGMGGI | SGMGCI |
| | 62 | G | C | GGITAV | GCITAV |
| PURA | 40 | G | R | GGGGSG | GRGGSG |
| TDP-43 | 290 | G | A | GFGNSR | GFANSR |
| | 290 | G | A | GNSRGG | ANSRGG |
| | 294 | G | A | GNSRGG | GNSRAG |
| | 294 | G | A | SRGGGA | SRAGGA |
| | 294 | G | V | NSRGGG | NSRVGG |
| | 294 | G | V | RGGGAG | RVGGAG |
| | 294 | G | V | GNSRGG | GNSRVG |
| | 295 | G | R | GGGAGL | GRGAGL |
| | 295 | G | S | GGGAGL | GSGAGL |
| | 298 | G | S | AGLGNN | ASLGNN |

Proteins like FUS, hnRNPA1, hnRNPA2, and TDP43 are known to form reversible phase separations, as well as irreversible amyloid aggregates. Numerous mutations occurring in the head domain of KRT8 were also identified, primarily associated with liver disease. Mutations in proteins GATAD1 (GATA zinc finger domain-containing 1) and PURA (transcriptional activator of Pur-alpha) were also identified by the in silico screen.

a fluorescent assay in which the sample is maintained at 37 °C, then cooled to 4 °C to induce phase separation, and then warmed back to 37 °C to melt the phase separations. Thioflavin T fluorescence occurs during phase separation at cool temperatures but diminishes when the sample is warmed (Fig. 3e). This effect has been previously demonstrated to occur for phase separation droplets of hnRNPA1 (ref. [27]). The fluorescent aggregates appear more amorphous than do some other condensates, perhaps owing to experimental variability, as the imaged sample was in the larger bulk volume of a plate well, not on a glass slide (see Methods). However, the amorphous appearance may result from the transition to a more solid or fibrillar aggregate state, not pure liquid-phase separation. Wild-type and mutant samples at 37 °C showed a low baseline of ThT fluorescence, but when cooled to 4 °C, all mutants, particularly G55A and Y54H, showed much more robust phase separation than did the wild-type protein (Fig. 3f). It then took several hours of melting at 37 °C to return to baseline. This same series of experiments was performed using FUS and three disease-related mutations (G191S, G225V, and G230C) that were identified in our screen (Extended Data Fig. 5). The results are like those for KRT8, in that each of the FUS mutants

reversibly aggregated much more extensively than did the wild type. Interestingly, the FUS phase separations took less than an hour to melt back to baseline, whereas KRT8 took nearly 5 hours.

**The effects of ethanol and the seeded aggregation of KRT8.** KRT8 is known to aggregate pathologically, primarily in the form of MDBs in liver disease. Given our structural and biochemical findings on KRT8 aggregation, as well as the fact that MDBs are most frequently observed in ASH and alcoholic cirrhosis, we next sought to explore KRT8 aggregation in the context of alcoholic liver disease[29]. Suggesting an effect of alcohol on KRT8 aggregation, two of three of our crystal structures of KRT8 segments showed the peptides in complex with either ethanol or isopropanol, each with distinct binding sites (Fig. 4a,b). The exact binding of the alcohols to the peptide segments seen in the crystal structures may not precisely represent the mode of interaction that alcohol has with the head domain/full-length protein in vivo. However, these structures provide initial insight into the KRT8–alcohol relationship and prompted us to further probe that relationship. To assess the effects of ethanol on KRT8 aggregation, wild-type $KRT8_{1-90}$ was aggregated in the presence of 0%, 1%, and 2.5% ethanol. Increasing concentrations of ethanol had a pronounced effect on promoting KRT8 aggregation (Fig. 4c). To see whether these effects were specific to amyloid aggregation of KRT8, we also assessed the aggregation of the amyloid proteins alpha-synuclein and tau in the presence of ethanol. Ethanol had a mild effect on the aggregation of alpha-synuclein (Fig. 4d) and no effect on tau (Fig. 4e). Substantial protein precipitation was observed for alpha-synuclein at 2.5% ethanol, thus results with only 0% and 1% ethanol are reported.

The in vitro aggregation of soluble amyloid-forming proteins can be seeded by the addition of tissue that contains amyloid aggregates. For example, aggregation of tau can be seeded by the addition of AD brain extracts containing neurofibrillary tangles, which are amyloid aggregates composed of tau[30]. To test the seeded aggregation of KRT8, we aggregated monomeric $KRT8_{1-90}$ in the presence of liver tissue extracts from people with either ASH or hepatocellular carcinoma (HCC) (Fig. 4f). There was no effect on KRT8 aggregation in the HCC samples, whereas two of three of the ASH samples had significantly enhanced aggregation. Western blot analysis of the four liver tissue extract samples tested shows dimerization of KRT8, which has been observed in previous analysis of MDB-containing liver tissues[31] (Extended Data Fig. 6). High-molecular-weight species of KRT8 are also seen in the samples, but interestingly their abundance does not correlate to seeding activity of the sample. Like in the extracts from people with ASH, we observed that KRT8 aggregation was also seeded by the addition of preformed recombinant aggregates (Extended Data Fig. 7). We next aimed to identify a compound that could mitigate KRT8 aggregation. A small screen of about 20 on-hand compounds, either known or predicted to reduce amyloid aggregation, including EGCG, curcumin, methylene blue, and others, identified the antibiotic demeclocycline HCl as capable of reducing KRT8 aggregation at even low nanomolar concentrations (Fig. 4g). Both demeclocycline and EGCG also reduced KRT8 phase separation (Extended Data Fig. 8).

## Discussion

By analyzing the conversion of LARKS into steric zippers, this work expands the understanding of the molecular mechanisms driving irreversible amyloid aggregation. LARKS are adhesive protein segments that promote mated beta-sheets, forming amyloid-like fibrils[6]. In contrast to pleated protein segments, kinks present in the protein backbone of LARKS limit the adhesion between the mated sheets, reducing the formation of steric zippers, which feature interdigitation of side chains protruding from the mating beta-sheets that stabilize pathogenic amyloid structures. Limited adhesion may lead to the observed reversibility of LARKS. LARKS are found in

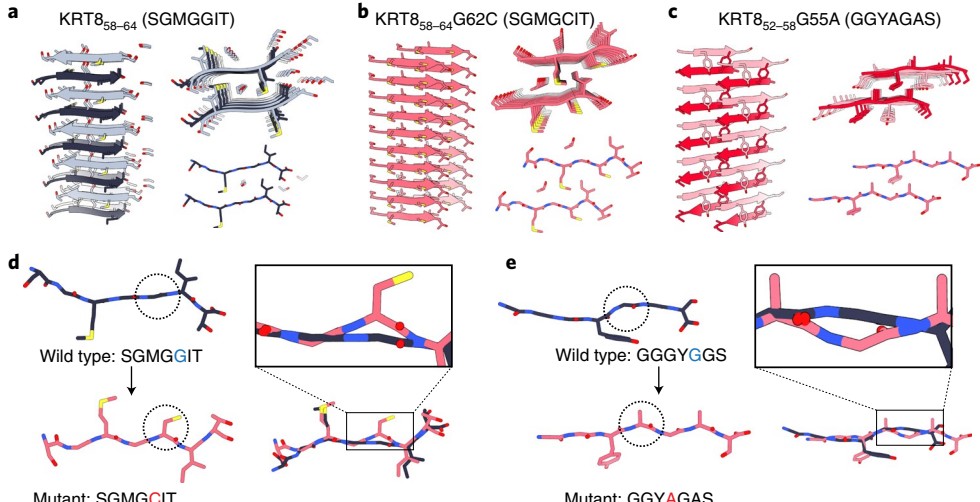

**Fig. 2 | Atomic structures of KRT8 amyloid segments.** Structures of LARKS regions from KRT8 that contain pathogenic mutations were crystalized to assess the effect that the mutations have on fibril structure. **a,b**, Fibril structures of the wild-type KRT8$_{58-64}$ segment SGMGGIT (**a**) and the corresponding G62C mutant structure SGMGCIT (**b**). Three views of the fibril structures are shown, from the side (left), looking down the fibril axis (top right), and showing a single fibril layer in cross-section (bottom right). The wild-type fibril contains an anti-parallel beta-sheet, with PEG and ethanol molecules bound along the fibril axis. The central glycine residues (Gly61 and Gly62) adopt a nearly linear beta-strand conformation. The G62C mutant fibril is a parallel beta-sheet, whose steric-zipper interface is driven by hydrophobic interactions between Met60, Cys62, and Ile63. **c**, The KRT8$_{52-58}$-G55A mutant structure GGYAGAS was also determined, revealing an anti-parallel beta-sheet with a steric-zipper interface formed by Tyr54, Ala55, and Ala57. **d**, Alignment of the wild-type and mutant KRT8$_{58-64}$ segments shows a transition in backbone conformation from a highly extended to a pleated beta-sheet at the site of the mutation (circled). Alignment of the beta-strands from both structures highlights that the G62C mutation results in the non-pleated Gly62 converting to a pleated conformation as Cys62 (shown in inset). **e**, Comparison of the KRT8$_{52-58}$-G55A GGYAGAS segment with a structure from hnRNPA1 whose sequences closely match the corresponding KRT8 wild-type (hnRNPA1: GGYGGS versus KRT8: GGYGGAS) shows a similar trend in transitioning from an extended beta-sheet in the wild type to a pleated conformation in the mutant. Similar to G62C, backbone overlay with the G55A mutation converts a non-pleated Gly55 into a pleated Ala55 (right).

many proteins known to undergo LLPS, including TDP-43 (ref. [32]), hnRNPA2 (ref. [11]), and FUS[10].

To deepen the understanding of the structural basis for disease-associated protein aggregation, our computational screen identified a set of mutations in LARKS. Further analysis suggests that these mutations tend to convert LARKS to steric zippers, thereby transitioning functional aggregation to pathogenic aggregation. In this analysis, we compared the relative frequencies at which the wild-type and mutant residues are found in beta-sheets in known protein structures (Extended Data Fig. 9a). Of eight mutations, two are to valine, which strongly encourages formation of beta-sheets, but other mutations do not have a strong trend of favoring beta-sheets. In contrast, we find a strong trend in the frequencies at which the wild-type residues are mutated from residues found in known LARKS (Extended Data Fig. 9b). That is, it appears that disruption of the LARKS conformation, notably at kinked glycine residues, could be a major factor in driving aggregation as opposed to simply adopting a more beta-sheet-prone sequence. Proline replacements, which are also known to introduce kinks into beta-sheets, might be expected to have a similar effect as glycine replacement.

Of structural interest, instead of highly kinked beta-sheets seen in the wild-type LARKS of FUS, TDP-43, hnRNPA1, and hnRNPA2, the glycine residues of the wild-type KRT8 segments assume a highly extended conformation, with both dihedral angles approaching |180°|. This extended conformation has also been observed in the structure of a LARKS from the nucleoporin protein Nup54 (ref. [33]). Additionally, evidence of these extended conformations can be found in the full-length fibril structures of FUS and TDP-43 (Extended Data Fig. 10). Both kinked and extended beta-sheet conformations may serve the same function, to limit or disrupt the pleated beta-sheet interdigitation that permits steric-zipper formation. Threading of sequences onto extended beta-sheet structures, as

we did to screen for LARKS, could be a useful future step to identify similar motifs that are important for regulating amyloid structure.

Our computational approach has increased the likelihood that alcoholic liver disease is an amyloid-related condition, in which KRT8 aggregation is associated with pathology. The same general procedure that led us to this conclusion could possibly be followed to discover other diseases in which amyloids are involved. First, assessing the propensity of sequences to adopt LARKS conformations can identify LCD-containing proteins that are able to engage in amyloid-like interactions. Of the ~1,700 proteins predicted to contain LARKS, many contain disease-related mutations in the LARKS sequence. In this study, we analyze only the 400 top-ranked LARKS (as scored by the structural threading algorithm), leaving many more to be screened and characterized[6]. We also focus exclusively on single missense mutations, but investigation of other LARKS and into other types of variants, such as deletions of translocations, may turn up other amyloid diseases[34]. Second, by predicting the impact that the mutations in LARKS have on potential steric-zipper formation, one can identify mutations that may impact protein aggregation. Expansion of this approach to include disease-associated mutations within entire LCDs, not just LARKS, may cast an even broader net to find more aggregation-promoting mutations. By identifying pathogenic mutations predicted to enhance steric-zipper formation, this workflow provides an approach to find proteins with a baseline propensity to aggregate in a disease-associated context.

Our analysis of KRT8 shows that each of the mutations predicted to promote steric-zipper formation accelerates aggregation, and confirms that KRT8 can undergo amyloid-like aggregation. Analysis of the sequence of the KRT8 head domain shows a hydrophobic segment in the low-complexity region, which may represent the potential fibril core. Comparison of this hydrophobic region of KRT8 with the amyloidogenic NACore of alpha-synuclein shows

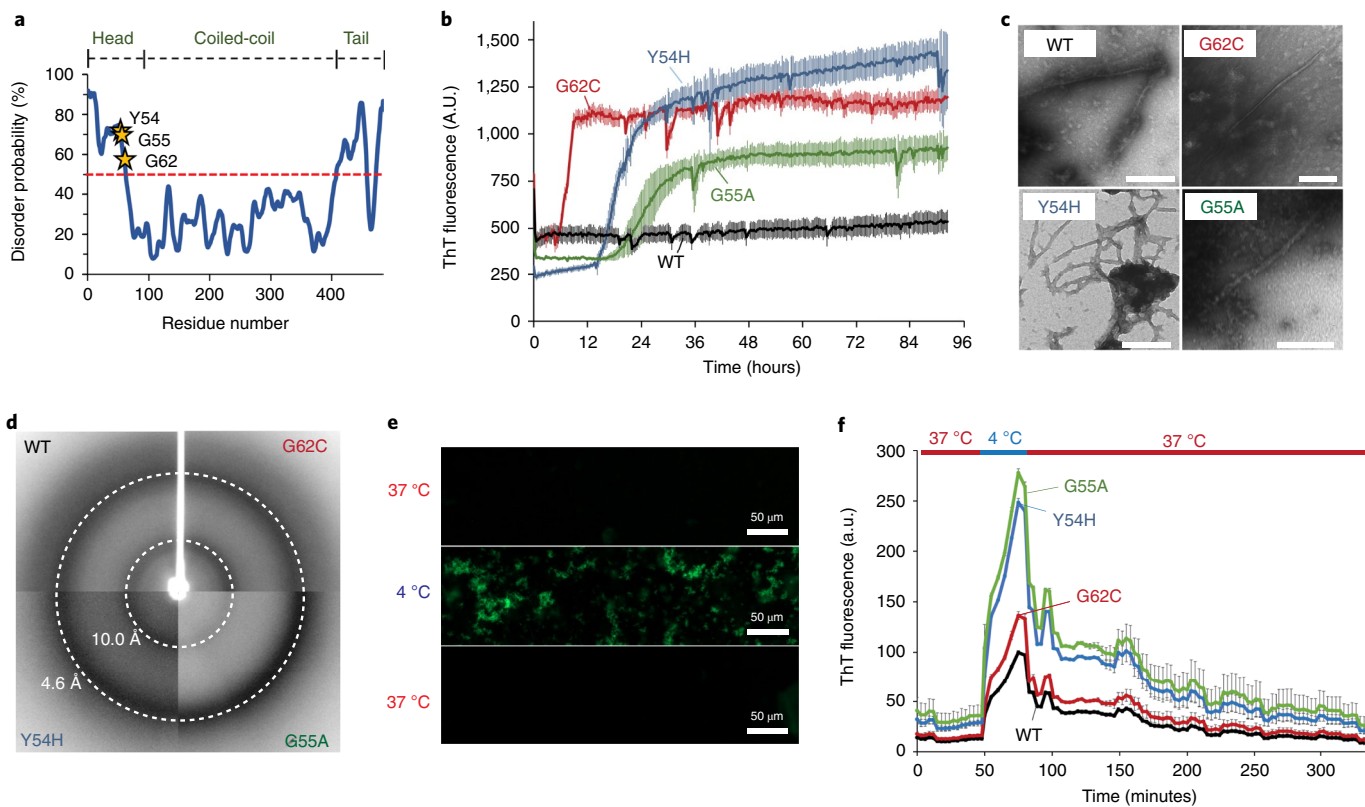

**Fig. 3 | The effects of mutations on the in vitro aggregation of KRT8. a**, Like other aggregation-prone low-complexity domains, the head domain of KRT8 is predicted to be disordered, and contains three disease-associated mutations, Y54H, G55A, and G62C. PrDOS prediction shows both the head and tail KRT8 domains, both of which have low-complexity sequence compositions, are above the threshold for predicted disorder (50%). **b**, Thioflavin T-based aggregation kinetics of KRT8 show that disease-related mutations present in LARKS sequences of the protein (Y54H, G55A, G62C) greatly enhance aggregation compared with wild type (WT). **c**, Electron micrographs of both wild-type and mutant KRT8 samples show fibrillar aggregates of varying morphology (scale bar, 400 nm). Very few fibrils were observed for the wild type, and many were seen for the three mutants. Representative micrographs were taken in triplicate for each experimental condition. **d**, X-ray powder diffraction of KRT8 aggregates produces a cross-beta diffraction pattern characteristic of amyloid fibrils. Diffraction rings are observed at 4.6 Å and 10 Å. **e**, Cooling of KRT8 from 37 °C to 4 °C produces ThT-positive aggregates, which dissociate upon warming back to 37 °C. **f**, Aggregation by cooling occurs more extensively in the mutant forms of KRT8 compared with the wild type, as assessed by ThT fluorescence. Samples were initially maintained at 37 °C, cooled to 4 °C, then melted back to 37 °C. All aggregates melt over the course of several hours after warming back to 37 °C. All aggregation assays were performed with $n = 3$ replicates. All error bars represent ±s.d.

a high degree of sequence similarity (Table 2). Work by McKnight and coworkers has demonstrated that the head domains of other intermediate filament proteins, desmin and neurofilament light, can drive amyloid aggregation, as well as the tail domain of intermediate filament protein Tm1 (refs. [16,17]). Their work also identifies several disease-causing mutations in desmin and neurofilament light that further potentiate amyloid aggregation.

Although not widely regarded as an amyloid protein, KRT8 is known to aggregate in liver disease within MDBs[35], and all three aggregation-promoting mutations analyzed in this study were initially associated with liver disease[26]. Composed primarily of aggregated KRT8 and lesser amounts of KRT18, MDBs are cytoplasmic inclusions that occur within hepatocytes most frequently in alcoholic steatohepatitis (ASH) or alcoholic cirrhosis[36]. Some have previously speculated about the amyloid nature of MDBs. MDBs have been demonstrated to bind the luminescent conjugated oligothiophenes h-HTAA and p-FTAA, both of which selectively bind proteins in cross-beta conformations[37]. Additionally, infrared spectroscopy shows cytokeratin transitioning from a predominantly helical conformation to predominantly beta-sheet when incorporated into MDBs[38]. Increased expression of keratins, as well as hyperphosphorylation, has been shown to occur during cellular stress, and the KRT8 mutations lead to visible KRT8 aggregation in cell culture models[26,39,40]. It is during these periods of increased

cellular concentration that factors like ethanol may further tip KRT8 to the point of aggregation.

We observe that KRT8 aggregation is enhanced by the presence of ethanol. The pro-aggregation effect of ethanol is also observed for alpha-synuclein, but not tau (Fig. 4c–e). A possible explanation for why KRT8 is sensitive to ethanol is that the KRT8 head domain contains a high concentration of glycine and polar residues, such as serine, relative to the amyloid core of tau, which is relatively hydrophobic. These polar residues may have increased interaction with the alcohols, somehow stabilizing their amyloid form. In vitro, wild-type KRT8 appears to form few solid aggregates alone, but once mutations or alcohol are introduced, aggregation is greatly enhanced. However, tau readily forms solid amyloid aggregates after only 1.5 hours in the presence of the aggregation-inducer heparin, much faster than does KRT8. The fact that tau is already so prone to aggregation in these conditions might preclude any drastic pro-aggregation effects seen following the addition of ethanol. This may also explain why increased aggregation with ethanol is seen for alpha-synuclein, which also has slow aggregation kinetics compared with tau. It should also be noted that, in the absence of heparin, tau is regarded as aggregation resistant compared with alpha-synuclein or amyloid-beta.

Just as the aggregation of other amyloid proteins can be seeded by preformed fibrils, we observe that KRT8 aggregation can be seeded by ASH liver tissue extracts. MDBs present in the livers of people

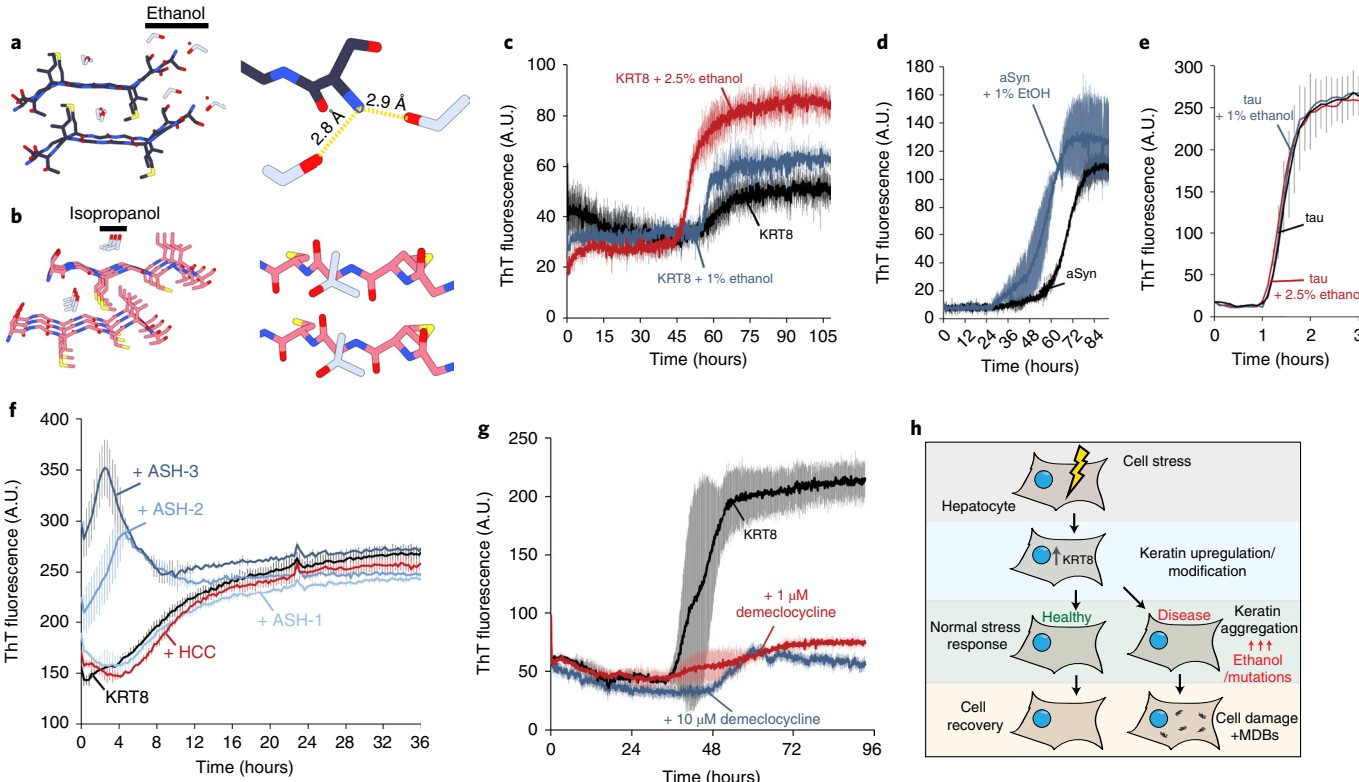

**Fig. 4 | Both ethanol and alcoholic steatohepatitis liver tissue extracts promote the thioflavin T-monitored aggregation of KRT8. a,b**, Structures of wild-type and mutant KRT8$_{58-64}$ segments co-crystallized with alcohol. The wild-type SGMGGIT structure (**a**) shows two ethanol molecules (gray) per asymmetric unit coordinated to the N terminus of the peptide, while mutant SGMGCIT (**b**) displays isopropanol binding alongside the middle of the peptide along the fibril axis, occupying the space between the beta-sheets (in gray) in the steric zipper. Both structures illustrate that alcohols can directly associate with KRT8 amyloid fibrils. Aggregated KRT8 is the primary component of MDBs, a common pathological feature observed in hepatocytes during alcoholic liver disease. **c**, ThT aggregation kinetics assay of wild-type KRT8 with increasing concentrations of ethanol (0, 1, and 2.5%). **d,e**, Aggregation of alpha-synuclein (**d**) and tau microtubule-binding domains (**e**) in the presence of ethanol. For alpha-synuclein ('aSyn'), significant protein precipitation was noted for the 2.5% ethanol sample, thus results from only the 1% ethanol sample are reported. **f**, Extracts of liver tissue with ASH or HCC were used to seed the aggregation of KRT8. **g**, The effects of demeclocycline HCl on KRT8 aggregation as measured by ThT fluorescence. We mixed 50 μM of KRT8$_{1-90}$ with various concentrations of demeclocycline dissolved in DMSO. The 'KRT8' control was treated with equal concentration DMSO vehicle. **h**, In a state of cellular stress, increased expression of KRT8 may contribute to a normal stress response. However, in the context of disease, particularly in the presence of ethanol and/or mutations which may occur in liver disease, KRT8 may become aggregated. This may result in the formation of MDBs or an insufficient stress response. $n = 3$ experimental replicates were used for **c–f**. Error bars represent ±s.d.

## Table 2 | Sequence comparison of KRT8$_{1-90}$ with alpha-synuclein

| Protein | Sequence |
|---|---|
| Alpha-synuclein | MDVFMKGLSKAKEGVVAAAEKTKQGVAEAAGK TKEGVLYVGSKTKEGVVHGVATVAEKTKE**QVTNV GGAVVTGVTAV**AQKTVEGAGSIAAATGFVKKDQL GKNEEGAPQEGILEDMPVDPDNEAYEMPSEEGY QDYEPEA |
| KRT8$_{1-90}$ | SIRVTQKSYKVSTSGPRAFSSRSYTSGPGSRISSSSFS RVGSSNFRGGLGGGYGGASGM**GGITAVTVNQ**SLL SPLVLEVDPNIQAVRTQ |

The head domain of KRT8 has an enrichment of glycine and serine residues. Additionally, it has a hydrophobic region (highlighted in red) that resembles the amyloidogenic core of alpha-synuclein, known as the NACore (highlighted in red).

with ASH reduce in size and quantity with cessation of alcohol consumption. However, if the patient returns to alcohol consumption, the MDBs return more rapidly and extensively than before[41]. Latent KRT8 aggregates may remain present during the alcohol cessation period and quickly seed the formation of subsequent aggregates,

contributing to what was originally described as the 'toxic memory' response of MDBs[36]. Given the limited therapeutic options available for the treatment of ASH and alcoholic cirrhosis, targeting KRT8 aggregation could present a new route for drug development, as has been done with other amyloid diseases. However, similar to other amyloid diseases, the causative versus correlative nature of KRT8 aggregates in relationship to liver disease progression requires further elucidation—whether these aggregates act as bystanders or directly contribute to pathogenesis merits further study[42].

Our principal findings are that a computational screen can identify diseases, previously unsuspected to be associated with protein aggregation, to be amyloid conditions, and that alcoholic liver disease is an example of such a disease. At present, there are more than 40 known amyloid diseases. The method presented here, exemplified by the evidence we present for the identification of alcoholic liver disease as amyloid-related, can be readily applied to uncover other previously unsuspected amyloid-related conditions.

## Online content

Any methods, additional references, Nature Research reporting summaries, source data, extended data, supplementary information, acknowledgements, peer review information; details of

author contributions and competing interests; and statements of data and code availability are available at https://doi.org/10.1038/s41594-022-00774-y.

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

## Methods

**Computational identification of LARKS mutants.** LARKS that are present in the human proteome were identified using the method by Hughes and colleagues[6]. We threaded protein sequences onto three LARKS backbone structures (STGGYS, SYSGYS, and GYNGFG), and the sequences were scored for their probability to form LARKS using the Rosetta energy function. Information about disease-related mutations was based on records from ClinVar[21], OMIM[19], and UniprotKB[20]. The records were retrieved in April 2018 with Python scripts utilizing NCBI E-utilities or custom APIs provided by OMIM and UniprotKB. SNP records were retrieved from the Entrez SNP database using NCBI E-utilities[43]. The UniprotKB mapping tool was used to convert UniprotKB accessions to Entrez Gene identifiers. Wild-type and mutant LARKS sequences were threaded onto steric-zipper backbones using ZipperDB[3]; energies of each were then compared and plotted.

**Crystallization.** All peptides used for crystallization were purchased from GenScript and crystals were grown in the following conditions, using hanging drop vapor diffusion: SGMGGIT: 0.1 M phosphate/citrate (pH 4.2), 40% vol/vol ethanol, 5 % wt/vol PEG 1000; SGMGCIT: 0.2 M sodium citrate tribasic dehydrate, 0.1 M HEPES (pH 7.5), 20% vol/vol 2-propanol; GGYAGAS: 2.5 M sodium chloride, 100 mM sodium acetate/acetic acid (pH 4.5), 0.2 M lithium sulfate.

**X-ray crystallography data collection and processing.** All X-ray diffraction data were collected using beamline 24-ID-E of the Advanced Photon Source (APS) at Argonne National Laboratory, using a beam wavelength of 0.971 Å and temperature of 100 K. Data were collected using 5° oscillations with a detector distance of 140 mm, using an ADSC Q315 CCD detector. Indexing and integration were performed using CCP4, and molecular replacement was done using Phaser, using a library of poly-alanine beta-strands, as well as a selection of kinked strands, as search models. Manual adjustments were performed with COOT during iterative rounds of processing with Refmac to refine the final atomic models. Data and refinement statistics are listed in Table 3.

**Protein expression and purification.** Recombinant $KRT8_{1-90}$ and $FUS_{1-214}$ were purified using a pHis-parallel-mCherry vector, using the method by Kato et al.[7]. The pHis vector contains an N-terminal mCherry fusion and a His-tag for purification. Bacterial cultures were grown with shaking at 225 r.p.m. at 37 °C to an optical density at 600 nm ($OD_{600}$) of 0.4–0.8. Protein expression was induced with 0.5 mM IPTG, and the cultures were cooled to 20 °C and left to shake overnight. The following day, cultures were centrifuged at 8,000g for 7 minutes to pellet cells. Cells were lysed by sonication for 3 minutes (3 seconds on–3 seconds off at 70% power) in lysis buffer (50 mM Tris-HCl (pH 7.5), 500 mM NaCl, 1% Triton X-100, 2 M guanidine hydrochloride, 0.4 mg/mL lysozyme, 20 mM β-mercaptoethanol (BME), HALT protease inhibitor cocktail). Lysate was then sonicated at 21,000g for 15 minutes. The supernatant was then run on a 10-mL Ni-NTA gravity column at 4 °C. The column was washed with 300 mL washing buffer 1 (20 mM Tris-HCl pH 7.5, 500 mM NaCl, 20 mM imidazole, 20 mM BME, 0.1 mM PMSF, 2 M guanidine hydrochloride). The resin was washed with 50 mL of wash buffer 2 (20 mM Tris-HCl pH 7.5, 500 mM NaCl, 20 mM imidazole, 20 mM BME, 0.1 mM PMSF, 2 M urea). Protein was eluted with elution buffer (20 mM Tris-HCl pH 7.5, 500 mM NaCl, 250 mM imidazole, 20 mM BME, 0.1 mM PMSF, 2 M urea). Protein was concentrated, and the concentration was measured using a NanoDrop (abs, 280 nm) and stored at −80 °C until used.

Recombinant tau microtubule-binding domain (k18) was expressed and purified using a pNG2 vector33 in BL21-Gold *Escherichia coli* cells. Bacteria were grown at 37 °C to $OD_{600} = 0.8$ in LB, and were then induced with IPTG (0.5 mM) for 3 hours, lysed by sonication in lysis buffer (MES buffer (pH 6.8) with 1 mM EDTA, 1 mM $MgCl_2$, 1 mM DTT, and HALT protease inhibitor) and NaCl was added to a final concentration of 500 mM. After boiling for 20 minutes, lysate was clarified with 21,000g centrifugation for 15 minutes, and then was dialyzed in dialysis buffer (20 mM MES buffer (pH 6.8) with 50 mM NaCl and 5 mM DTT). Lysate was purified using a HighTrap SP ion exchange column and eluted over a NaCl gradient from 50 to 550 mM. Proteins were then run on a HiLoad 16/600 Superdex 75 pg in 10 mM Tris (pH 7.6), 100 mM NaCl, and 1 mM DTT, then concentrated to ~20–60 mg/ml using a 3-kDa cutoff by ultrafiltration spin column[44].

Recombinant α-synuclein was performed using BL21(DE3) Gold *E. coli* cells. Bacteria were grown at 37 °C to $OD_{600} = 0.8$ in LB, then induced with IPTG (0.5 mM) for 3 hours, lysed by sonication in lysis buffer (100 mM Tris- HCl pH 8.0, 1 mM EDTA pH 8.0). Lysate was clarified with 21,000g centrifugation for 15 minutes. Following addition of 10 mg/ml streptomycin, the supernatant was stirred for 30 minutes, then centrifuged at 21,000g for 30 minutes. The supernatant was discarded, and the pellet was resuspended in 20 mM Tris pH 8.0. The solution was then dialyzed in 20 mM Tris pH 8.0 overnight. Protein was then purified using HiPrep Q HP column (GE Healthcare) with buffer A (20 mM Tris pH 8.0) and buffer B (20 mM Tris pH 8.0; 0.5 M NaCl) on a gradient from 0% to 100% buffer B over 100 mL. Fractions were collected injected on a preparative size exclusion silica G3000 column (Tosoh Bioscience), using buffer of 0.1 M sodium sulfate, 25 mM sodium phosphate, and 1 mM sodium azide, pH 6.5. Protein was then dialyzed in 0.1 M sodium sulfate, 25 mM sodium phosphate twice. Protein fractions were collected and concentrated[45].

**In vitro aggregation assay.** Frozen aliquots of wild-type and mutant $KRT8_{1-90}$ were thawed on ice and diluted to 100 μM in buffer containing 20 mM Tris-HCl (pH 7.5), 200 mM NaCl, 0.5 mM EDTA, 50 μM ThT, and 5 μM TCEP to a final volume of 200 μL in black Nunc 96-well optical bottom plates (Thermo Fisher Scientific). A single PTFE bead (0.125-inch diameter) was added to each well to facilitate agitation. Plates were incubated in a microplate reader (FLUOstar OMEGA, BMG Labtech) for ~100 hours at 37 °C with 700 r.p.m. double orbital shaking. Fluorescent measurements were recorded every 15 minutes, with excitation wavelength $\lambda_{ex} = 440$ nm and emission wavelength $\lambda_{em} = 480$ nm.

For reversible aggregation assays, the same sample concentrations were used. Sample was maintained at 37 °C in the microplate reader, with 700 r.p.m. double orbital shaking. Readings were taken every 5 minutes. Once the ThT levels were at a baseline, the plate was transferred to a Torrey Pines orbital plate shaker maintained at 4 °C, with orbital shaking set at maximum. The plate was manually and briefly transferred back to the microplate reader to take fluorescence measurements every 5 minutes. After 60 minutes, the plate was then transferred back to the microplate reader and kept there to measure melting, with readings again taken every 5 minutes.

Liver tissue seeded assays were carried out in 1× PBS with 100 μM $KRT8_{1-90}$ and 50 μM ThT. Liver extract was in a final dilution of 1/100th from the stock extract. The sample was agitated with a PTFE bead with double orbital shaking at 700 r.p.m.

Both tau k18 and alpha-synuclein aggregation assays were performed in 1× PBS with 50 μM protein monomer and 50 μM ThT, and were agitated with a PTFE bead with double orbital shaking at 700 r.p.m.

For small-molecule inhibition assays, 100 mM DMSO stock for each compound was diluted with 1× PBS into a 1 mM working stock. Compounds were combined with 50 μM KRT8 and 50 μM ThT in 1× PBS and agitated with a PTFE bead with double orbital shaking at 700 r.p.m. KRT8 controls were matched with the same DMSO concentration as treated samples (final DMSO concentration, 0.1%).

All aggregation experiments were performed with $n = 3$ experimental replicates, with measurements taken from distinct samples.

**Table 3 | Table of peptide microcrystal data collection and refinement statistics**

|  | [58]SGMGGIT[64] | [58]SGMGCIT[64] | [52]GGYAGAS[58] |
| --- | --- | --- | --- |
| **Data collection** | | | |
| Space group | P2₁ | P2₁ | P1 |
| Cell dimensions | | | |
| a, b, c (Å) | 8.36, 51.61, 9.53 | 4.75, 46.17, 10.33 | 9.43, 10.49, 16.63 |
| α, β, γ (°) | 90.0,109.1,90.0 | 90.0, 103.3, 90.0 | 88.9, 76.3, 74.1 |
| Resolution (Å) | 1.1 | 1.7 | 1.1 |
| $R_{sym}$ or $R_{merge}$ | 0.182 (0.351) | 0.192 (0.947) | 0.059 (0.109) |
| $I / \sigma I$ | 3.3 (1.4) | 13.0 (2.1) | 7.43 (3.93) |
| Completeness (%) | 93.3 (62.7) | 93.5 (71.4) | 72.0 (50.7) |
| Redundancy* | 2.6 (2.0) | 4.4 (2.6) | 1.5 (1.5) |
| **Refinement** | | | |
| Resolution (Å) | 1.1 | 1.7 | 1.1 |
| No. reflections | 7,713 | 2,087 | 3,928 |
| $R_{work} / R_{free}$ | 0.157 / 0.169 | 0.291 / 0.269 | 0.280 / 0.306 |
| No. atoms | | | |
| Protein | 98 | 43 | 41 |
| Ligand/ion | 19 | 4 | 0 |
| Water | N/A | N/A | N/A |
| B-factors | | | |
| Protein | 9.1 | 30.1 | 7.1 |
| Ligand/ion | 18.0 | 34.1 | N/A |
| Water | N/A | N/A | N/A |
| R.m.s. deviations | | | |
| Bond lengths (Å) | 0.010 | 0.022 | 0.001 |
| Bond angles (°) | 1.5 | 2.0 | 0.6 |

*Single crystals were used for structure determination of SGMGGIT and GGYAGAS. Data from two crystals were merged for SGMGCIT. Values in parentheses are for highest-resolution shell.

**Transmission electron microscopy.** Six microliters of aggregated wild-type and mutant $KRT8_{1-90}$ samples (taken from in vitro aggregation experiments) was spotted onto Formvar Carbon film 400 mesh copper grids (Electron Microscopy Sciences) and incubated for 4 minutes. Grids were stained with 6 μL uranyl acetate solution (2% wt/vol in water) for 2 minutes. Excess solution was removed by blotting, followed by air drying for 30 minutes. Transmission electron microscopy (TEM) images were acquired with a JOEL 100CX TEM electron microscope at 100 kV.

**X-ray fiber powder diffraction.** Aggregated samples of KRT8 or FUS were centrifuged at 21,000$g$ for 30 minutes, and buffer was exchanged with water twice. Samples were suspended between two siliconated glass capillaries that were ~1 mm apart, forming a bridge between the two capillaries. The sample was allowed to dry, and the capillary with the dried aggregate was mounted on an in-house X-ray diffraction machine and diffracted with X-rays for 5 minutes, with the diffraction pattern collected on a CCD detector.

**SDS denaturation experiments.** KRT8 and FUS samples were aggregated in 1× PBS, 50 μM ThT for ~5 days in a FLUOstar OMEGA microplate reader (BMG Labtech) with 700 r.p.m. double orbital shaking in a Nunc 96-well optical bottom plate. Samples were then incubated with, with using $\lambda_{ex} = 440$ nm and $\lambda_{em} = 480$ nm, using the microplate reader. Experiments were performed with $n = 3$ experimental replicates, with measurements taken from distinct samples.

**Extraction of human liver tissue samples.** Tissues from people with histopathologically confirmed liver disease, either ASH or hepatocellular carcinoma, were fresh-frozen and extracted without freeze–thaw. Extraction of protein aggregates from these liver tissues was performed using a modified protocol from McGee and colleagues[46]. First, 250 mg of frozen tissue was cut and thawed, and then was homogenized using a manual tissue homogenizer. The samples were diluted fivefold in extraction solution (0.25 M sucrose (pH 8.5), 5 mM Tris-HCl, and 5 mM EDTA) on ice. Samples were then centrifuged at 4 °C at 4,000$g$ for 20 minutes. Supernatant was discarded, and the pellet was resuspended in the extraction solution and respun for an additional 20 minutes. Washes were repeated until the layer of floating lipid present after each round of centrifugation was no longer present. The solution was then passed through a 40-μm cell strainer to remove any intact debris. The filtrate was spun for 15 minutes at 400$g$, and the pellet was washed twice. The pellet was then resuspended in 1× PBS for use in seeding experiments.

**Western blot of human liver tissue samples.** Four separate human tissue samples were analyzed (ASH-1, ASH-1, ASH-3, HCC). An iBLOT2 dry blotting system was used to transfer protein from the gel to a nitrocellulose membrane. Membrane was blocked with 5% milk in TBST for 1 hour, then washed 3 times with TBST. The membrane was incubated with the primary antibody (Anti-Cytokeratin 8 Antibody (C51); sc-8020, Santa Cruz Biotechnology) at a 1:500 dilution in 2% milk/TBST solution) for 3 hours, washed 3 times with TBST, incubated with the horseradish-peroxidase-conjugated secondary antibody (goat anti-mouse IgG H and L (HRP); 1:1,000 dilution in 2% milk/TBST), and washed three times in TBST. The signal was detected with Pierce ECL Plus Western Blotting Substrate (cat. no. 32132), and imaging was performed with a Pharos FX Plus Molecular Imager. Actin was subsequently measured using the same protocol with a β-actin (C4) primary antibody (Santa Cruz Biotechnology) (1:250 dilution, 2% milk/TBST).

**Reporting summary.** Further information on research design is available in the Nature Research Reporting Summary linked to this article.

## Data availability

The datasets generated and/or analyzed during the current study are provided as source data. Any additional data are available from the corresponding author. All structural data have been deposited into the Worldwide Protein Data Bank (wwPDB) with the following accession codes: PDB 7K3C for SGMGGIT, PDB 7K3X for SGMGCIT, and PDB 7K3Y for GGYAGAS.

Calculations of zipper and LARKS propensity can be performed using our online web servers, which can be found at https://services.mbi.ucla.edu/zipperdb/ and https://srv.mbi.ucla.edu/LARKSdb/
Atomic coordinates for KRT8 segments SGMGGIT, SGMGCIT, and GGYAGAS have been deposited in the Protein Data Bank under accession codes 7K3C, 7K3X, and 7K3Y, respectively. Source data are provided with this paper.

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

## Acknowledgements

We thank S. French Sr. for advice regarding tissue extraction of KRT8, J. Treanor and E. Marcus for discussions, D. Cascio and M. Collazo in the UCLA Department of Energy Institute Macromolecular Crystallization Core Technology Center for assistance in structure determination, and the UCLA Translational Pathology Core Laboratory for their assistance in acquiring tissue samples. We acknowledge USPHS National Research Service Award 5T32GM008496 (K. A. M.), the UCLA-Caltech Medical Scientist Training Program (K. A. M.), and HHMI for support. L. S. is supported by NIH GM123126. This work is based on research conducted at the Northeastern Collaborative Access Team beamlines, which are funded by the National Institute of General Medical Sciences from the National Institutes of Health (P30 GM124165). The Eiger 16M detector on the 24-ID-E beamline is funded by a NIH-ORIP HEI grant (S10OD021527). This research used resources of the Advanced Photon Source, a US Department of Energy (DOE) Office of Science User Facility operated for the DOE Office of Science by Argonne National Laboratory under contract no. DE-AC02-06CH11357. We acknowledge support from NIH AG 054022, and AG 048120 and DOE DE-FC02-02ER63421 (D. S. E.).

## Author contributions

The project was conceived and designed by K. A. M. and D. S. Identification of LARKS in the proteome was done by M. P. H., identification of disease mutations within LARKS was done by L. S., and assessment of steric-zipper propensity for each mutation was measured by K. A. M. Crystallization and determination of KRT8 segment structures were performed by K. A. M. with assistance from M. R. S. Protein purification was performed by K. A. M. and C. J. H. Aggregation experiments, X-ray diffraction and SDS denaturation was performed by K. A. M. Liver extraction was performed by K. A. M. H. P. performed western blot analysis. Liver disease tissue samples were acquired and characterized with assistance from S. W. F. Electron microscopy was performed by P. M. S. and K. A. M. The manuscript was written by K. A. M. and D. S. E., with contributions from all other authors.

## Competing interests

D. S. E. is SAB chair and equity holder in ADRx, Inc. The remaining authors declare no competing interests.

## Additional information

**Extended data** is available for this paper at https://doi.org/10.1038/s41594-022-00774-y.

**Correspondence and requests for materials** should be addressed to David S. Eisenberg.

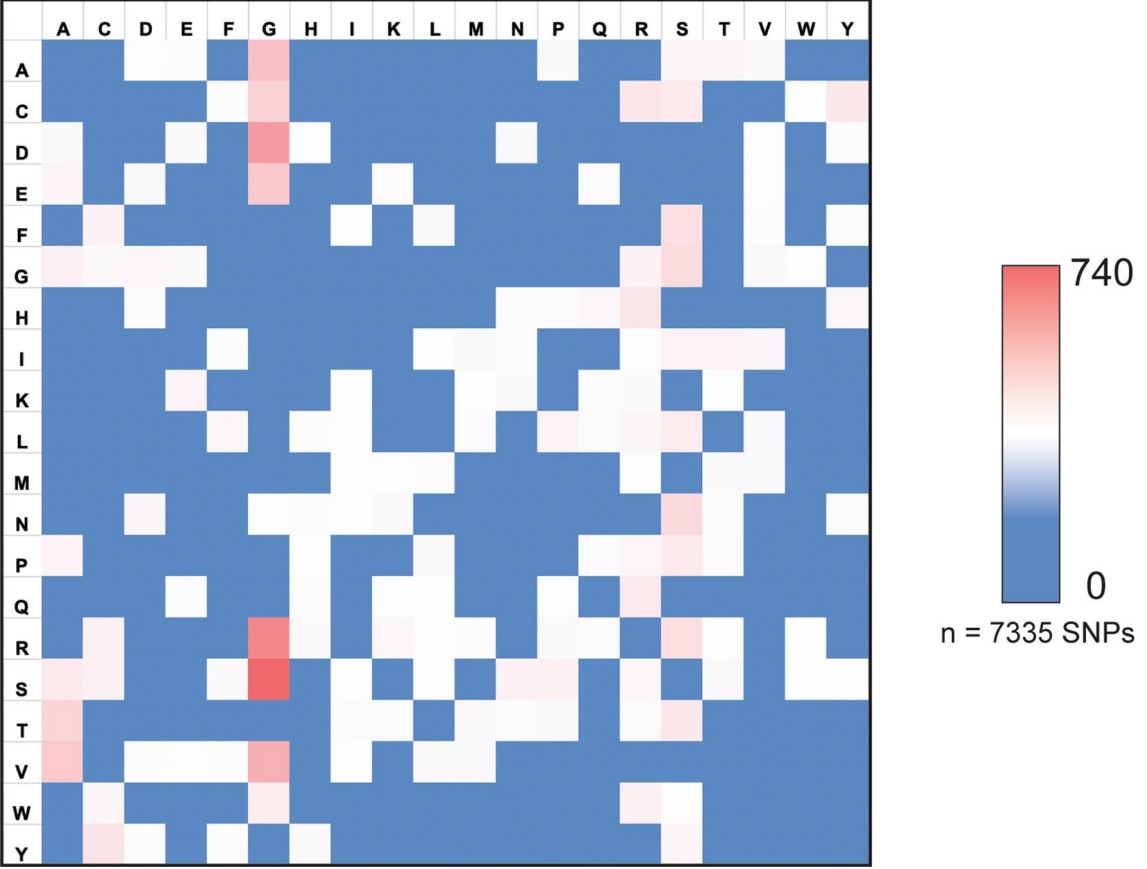

**Extended Data Fig. 1 | Heatmap for benign SNPs found in LARKS.** Wild type residues on shown on the y-axis, and mutant amino acids on the x-axis. Blue indicates low frequency; red indicates high frequency. The most commonly observed polymorphisms are those which lead to glycine, such as S→G. The benign nature of these variants may be due to the non-disruptive nature of the amino acid substitution. Glycine and serine residues are highly enriched in LARKS and low-complexity domains. The G→S and S→G variants may indicate a degree of interchangeability between glycine and serine residues is tolerated for function to be maintained. Data for heatmap is available as source data.

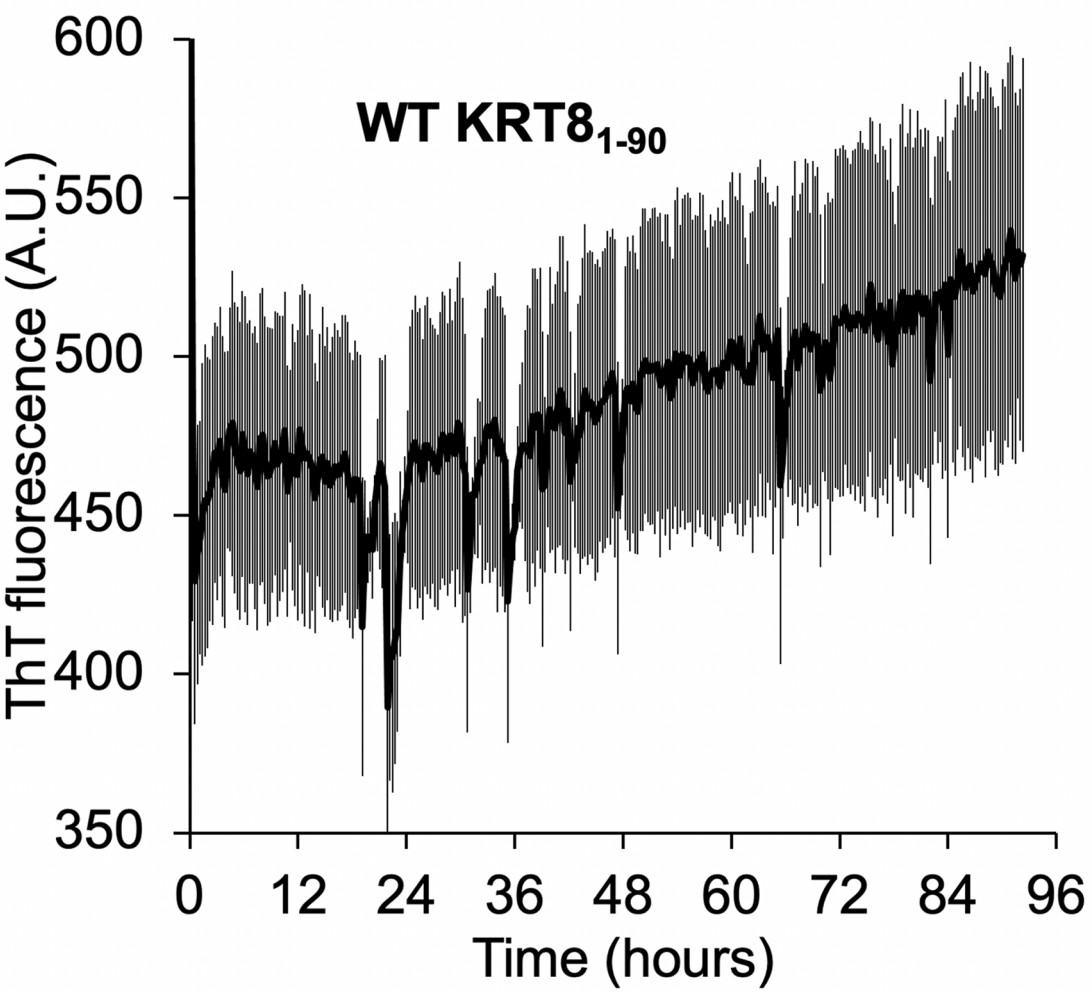

**Extended Data Fig. 2 | Thioflavin T aggregation kinetics assay of WT KRT8$_{1-90}$ aggregation.** View of WT KRT8$_{1-90}$ from Fig. 3b from main text with zoomed y-axis. ThT fluorescence appears to increase linearly following ~24 h. Assay performed with n = 3 replicates. Error bars represent ±SD.

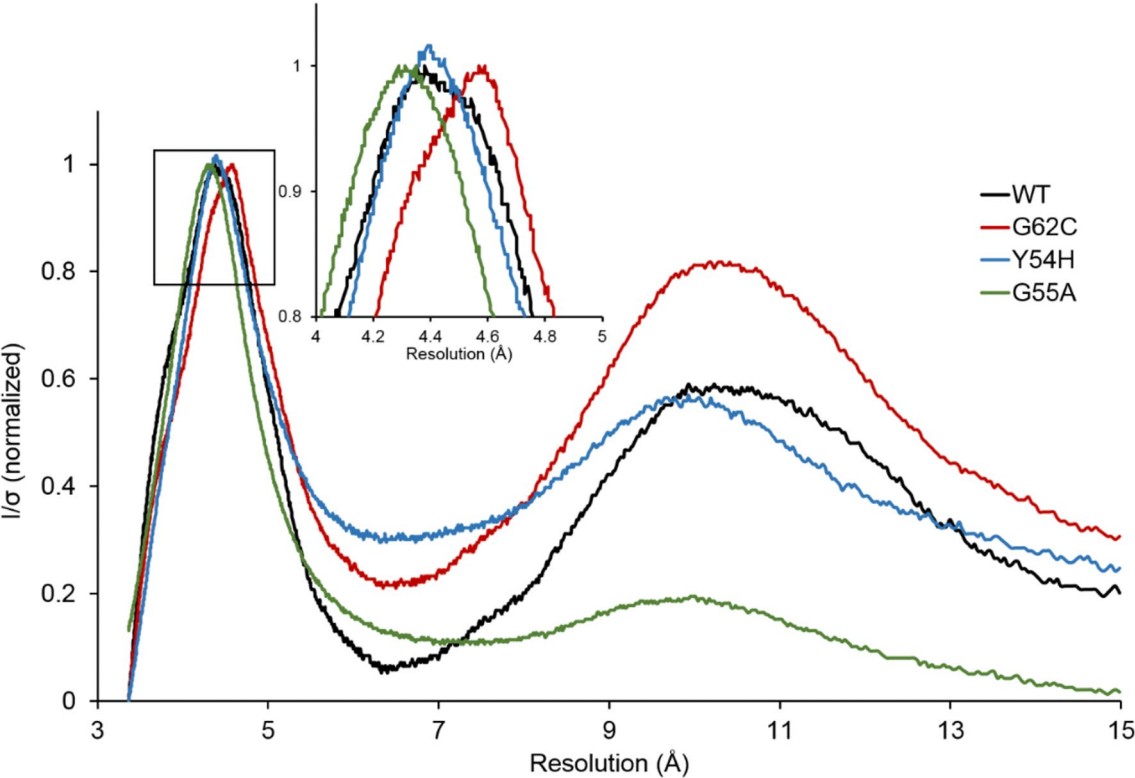

**Extended Data Fig. 3 | Radial profiles of x-ray powder diffraction of KRT8 aggregates.** A slight difference between the wild-type and mutants forms of KRT8 aggregates can be observed between the rings ranging 4.3-4.6 Å. The G55A mutant also produces a much weaker ring near 10 Å compared to the others. Data for graphs is available as source data.

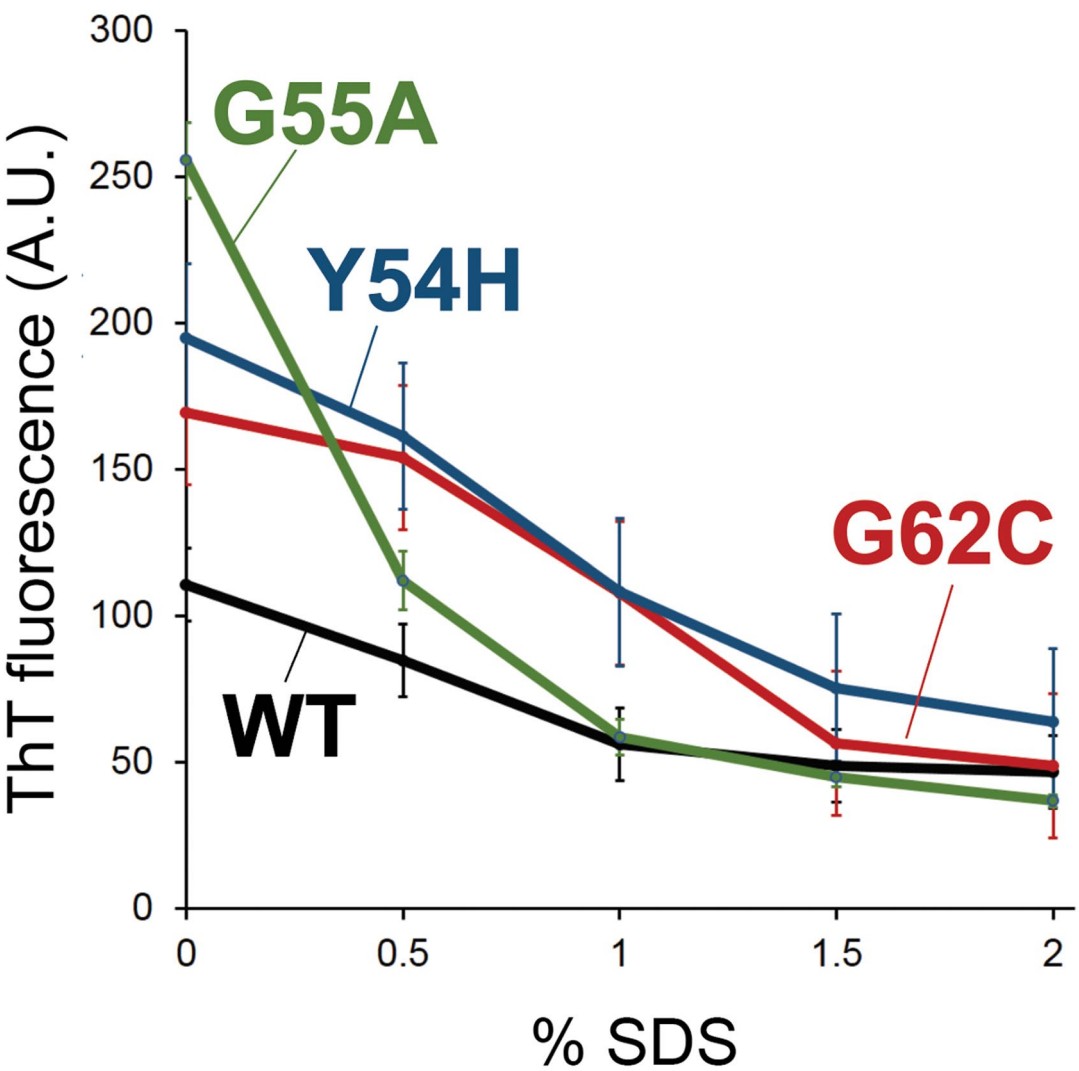

**Extended Data Fig. 4 | Stability of the wild-type and mutant KRT8 aggregates to SDS denaturation.** At 0% SDS, wild type KRT8 forms less fibrils overall compared to all three mutants, as measured by ThT fluorescence. Experiment performed with n=3 replicates. Error bars represent ± SD. Data for graph is available as source data.

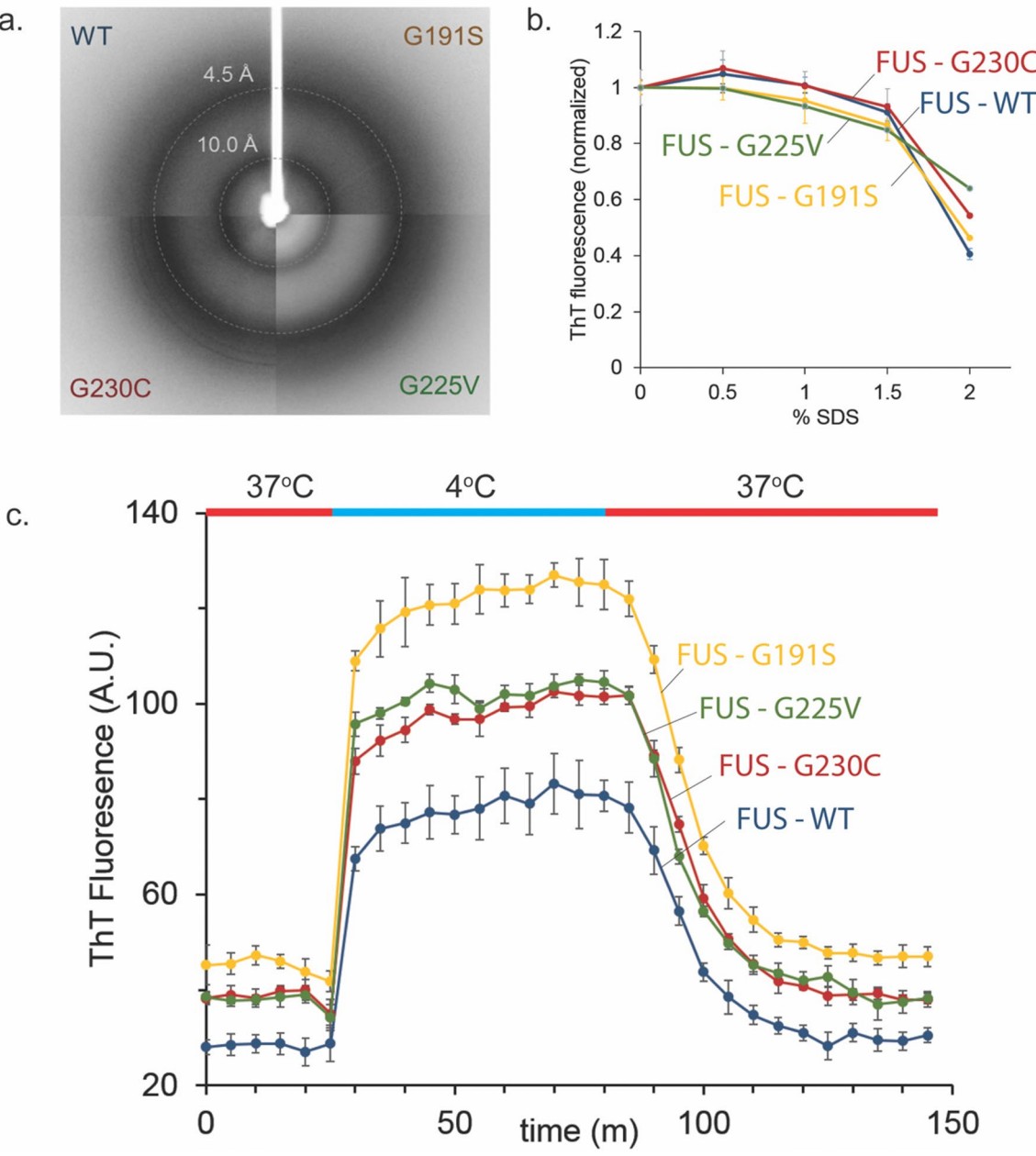

**Extended Data Fig. 5 | The effects of pathogenic LARKS mutations on FUS aggregation.** Aggregates of both wild-type and mutant (G191S, G230C, and G225V) FUS all produce cross-beta diffraction indicative of amyloid formation. **b**. FUS aggregates display similar stabilities to SDS denaturation, with the wild-type being slightly less stable at 2% SDS compared to the mutants. **c**. Like KRT8, FUS forms ThT positive phase separations when cooled to 4°. Experiment performed with n=3 replicates. Error bars represent ± SD. Data for graphs is available as source data.

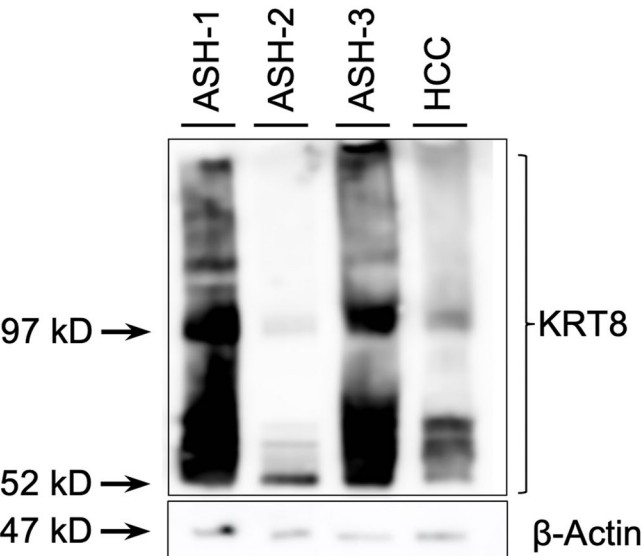

**Extended Data Fig. 6 | KRT8 Western blots of patient-derived liver tissue extracts for alcoholic steatohepatitis (ASH) and hepatocellular carcinoma.** Western blot analysis of the four liver tissue extracts used for seeding experiments shows monomeric KRT8 (52 KDa), dimerized KRT8 (~97 KDa), and smearing of other high molecular weight KRT8 species. Beta-actin was used as loading control, shown at the bottom panel. Western blots were performed in triplicate. Uncropped image is available as source data.

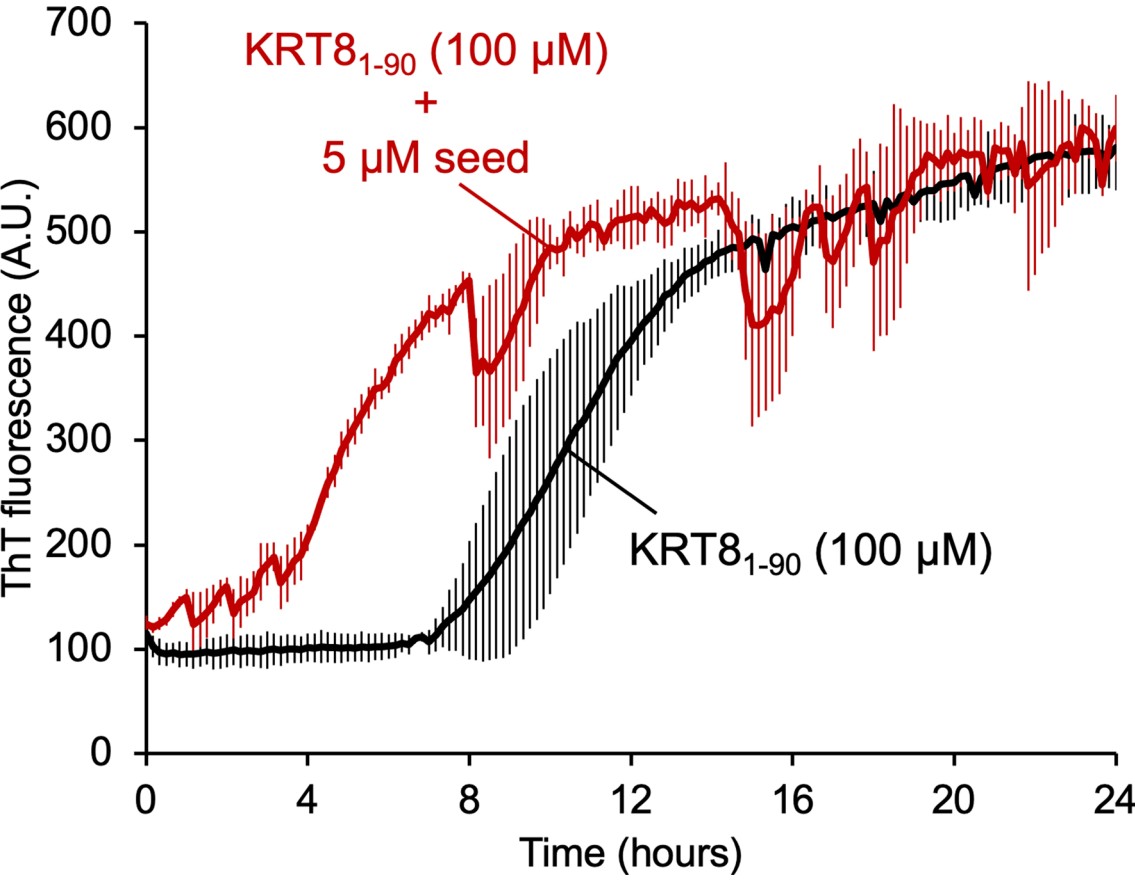

**Extended Data Fig. 7 | Seeded aggregation of KRT8 by preformed recombinant KRT8 aggregates.** Pre-aggregated KRT8$_{1-90}$ was added to monomeric KRT8$_{1-90}$ (100 μM) and allowed to aggregate. The addition of pre-aggregated KRT8 enhances the aggregation of monomeric protein, presumably through template seeding. Experiment performed with n = 3 replicates. Error bars represent ± SD. Data for graphs is available as source data.

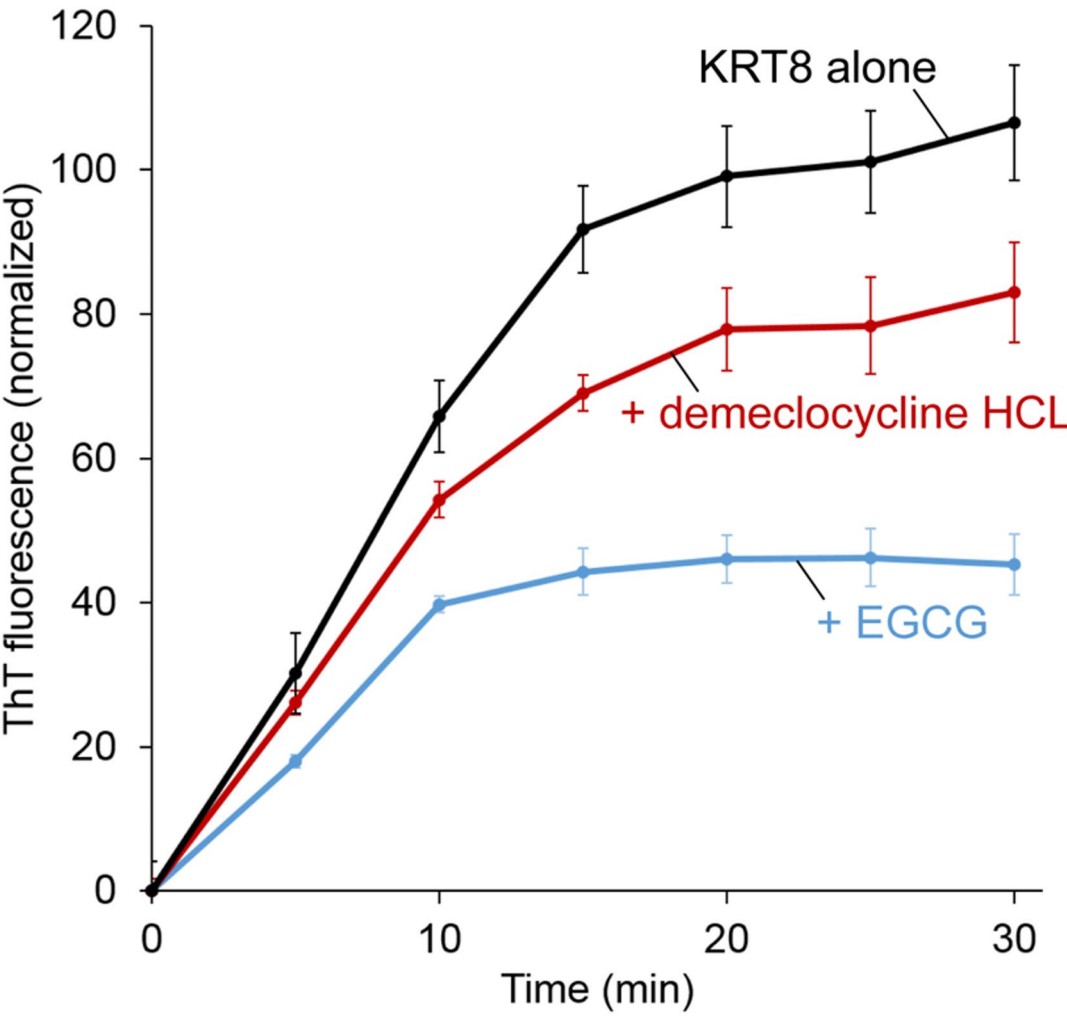

**Extended Data Fig. 8 | The effects of small molecules on KRT8 phase separation.** KRT8 samples were maintained at 37 °C at time point 0, then cooled to 4 °C, leading to ThT phase separations. At 50 μM KRT8 concentration, addition of demeclocycline HCl (10 μM) or the known anti-amyloid compound EGCG (10 μM) reduces ThT signal. Experiment performed with n = 3 replicates. Error bars represent ± SD. Data for graph is available as source data.

## Beta-sheet residue propensity

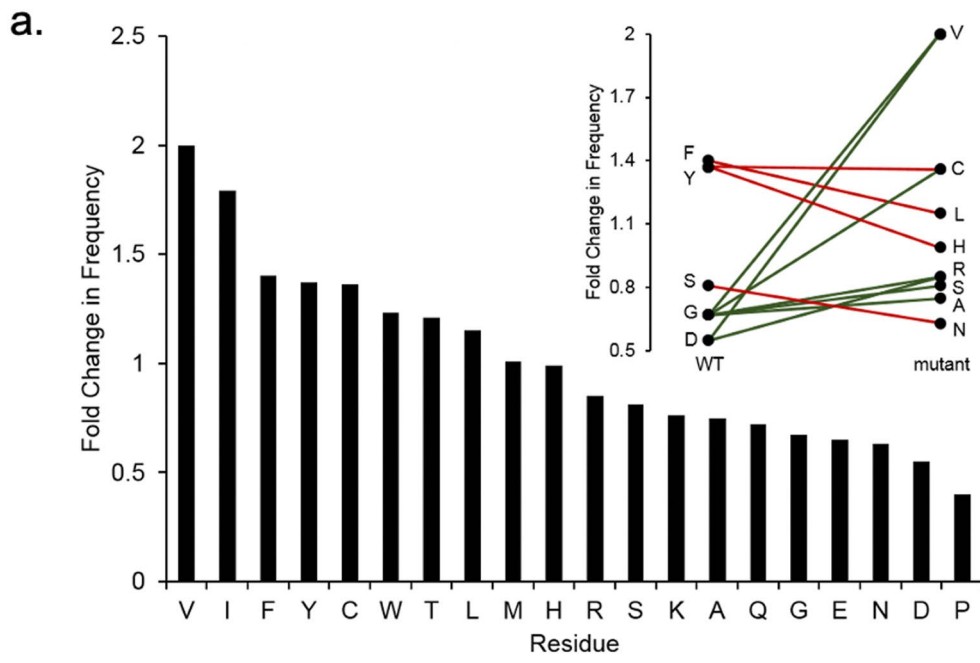

## LARKS residue propensity

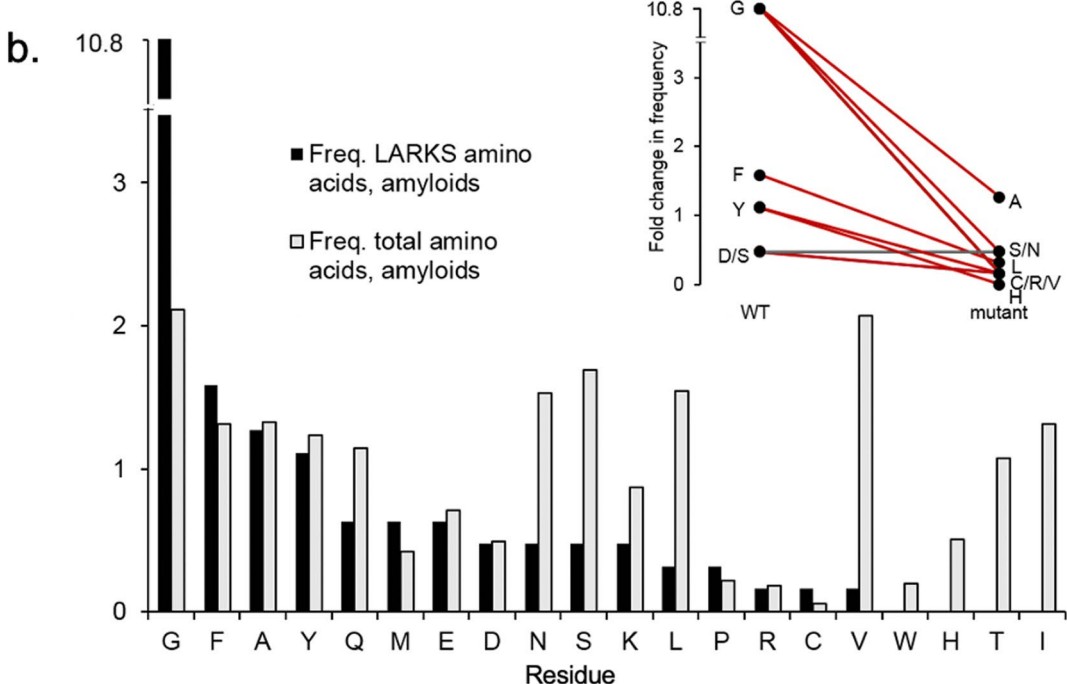

**Extended Data Fig. 9 | Residue frequencies in beta-sheets and LARKS. a**. The relative frequencies in which each amino acid is observed in beta-sheets (bar graph). Analysis of the disease-related mutations predicted to convert LARKS to zippers shows little correlation of residues mutating to more beta-sheet prone amino acids compared to wild-type (insert). **b**. Comparison of relative amino acid frequencies between all experimentally determined structures of LARKS versus amyloid fibrils (bar graph). There is a drastically higher enrichment of glycines in LARKS compared to canonical amyloid structures, likely contributing to their kinked structures. Pathogenic mutations predicted to convert LARKS to steric zippers lead to a transition from residues enriched in LARKS to those that are not. Data for graphs is available as source data.

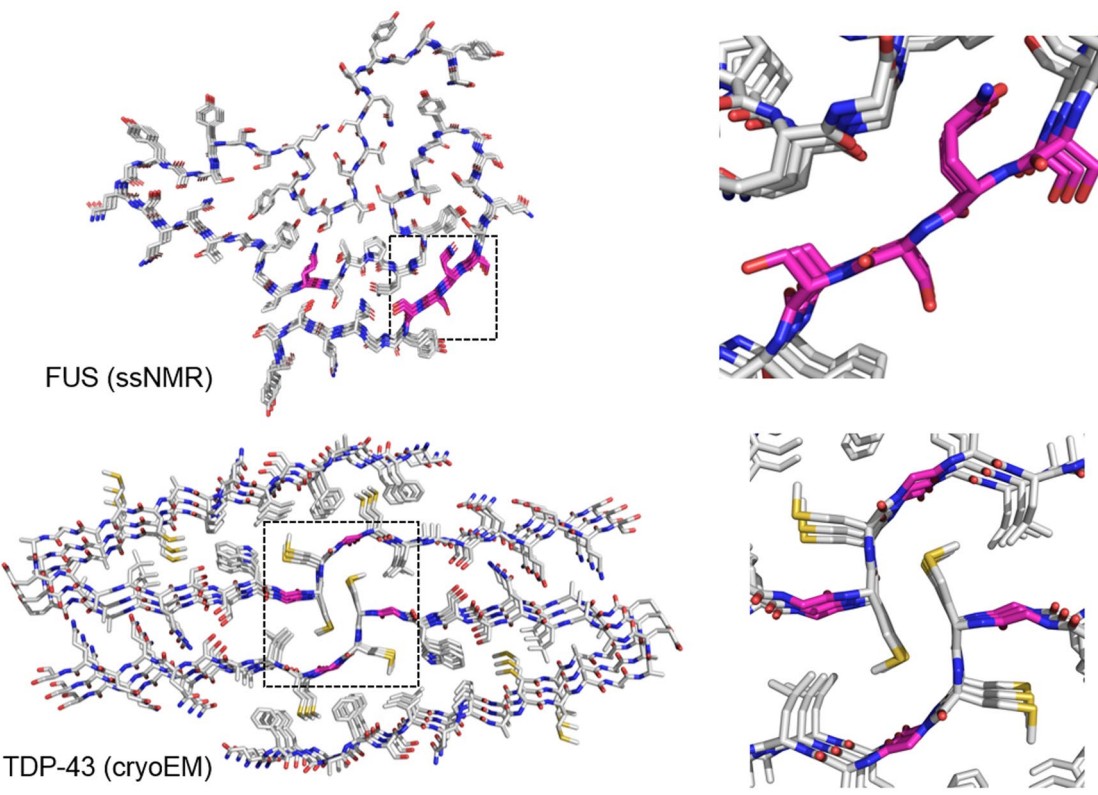

FUS (ssNMR)

TDP-43 (cryoEM)

**Extended Data Fig. 10 | Atomic structures of FUS and TDP-43.** Both structures have regions of highly extended beta-sheets (magenta).

# Reporting Summary

## Statistics

For all statistical analyses, confirm that the following items are present in the figure legend, table legend, main text, or Methods section.

| n/a | Confirmed | |
|---|---|---|
| ☐ | ☒ | The exact sample size (*n*) for each experimental group/condition, given as a discrete number and unit of measurement |
| ☐ | ☒ | A statement on whether measurements were taken from distinct samples or whether the same sample was measured repeatedly |
| ☒ | ☐ | The statistical test(s) used AND whether they are one- or two-sided<br>*Only common tests should be described solely by name; describe more complex techniques in the Methods section.* |
| ☐ | ☒ | A description of all covariates tested |
| ☒ | ☐ | A description of any assumptions or corrections, such as tests of normality and adjustment for multiple comparisons |
| ☐ | ☒ | A full description of the statistical parameters including central tendency (e.g. means) or other basic estimates (e.g. regression coefficient) AND variation (e.g. standard deviation) or associated estimates of uncertainty (e.g. confidence intervals) |
| ☒ | ☐ | For null hypothesis testing, the test statistic (e.g. *F*, *t*, *r*) with confidence intervals, effect sizes, degrees of freedom and *P* value noted<br>*Give P values as exact values whenever suitable.* |
| ☒ | ☐ | For Bayesian analysis, information on the choice of priors and Markov chain Monte Carlo settings |
| ☒ | ☐ | For hierarchical and complex designs, identification of the appropriate level for tests and full reporting of outcomes |
| ☒ | ☐ | Estimates of effect sizes (e.g. Cohen's *d*, Pearson's *r*), indicating how they were calculated |

*Our web collection on statistics for biologists contains articles on many of the points above.*

## Software and code

Policy information about availability of computer code

| Data collection | Zipper and LARKS score calculations were performed using publicly available code found on ZipperDB and LARKSdb, as detailed in the Data Availability section |
|---|---|
| Data analysis | GraphPad Prism (version 9.1.2) and Microsoft Excel for Mac (version 16.58) were used for all data analysis and statistics. Crystal structures were presented using UCSF Chimera (version 1.13.1). Atomic models were built in COOT (version 0.8.9.1). Model refinement was performed with phenix.real_space_refine (version 1.13-2998). |

For manuscripts utilizing custom algorithms or software that are central to the research but not yet described in published literature, software must be made available to editors and reviewers. We strongly encourage code deposition in a community repository (e.g. GitHub). See the Nature Portfolio guidelines for submitting code & software for further information.

## Data

Policy information about availability of data

All manuscripts must include a data availability statement. This statement should provide the following information, where applicable:
- Accession codes, unique identifiers, or web links for publicly available datasets
- A description of any restrictions on data availability
- For clinical datasets or third party data, please ensure that the statement adheres to our policy

The datasets generated and/or analyzed during the current study are provided as Source Data. Any additional data are available from the corresponding author.

All structural data have been deposited into the Worldwide Protein Data Bank (wwPDB) with the following accession codes: SGMGGIT (PDB 7K3C), SGMGCIT (PDB 7K3X), and GGYAGAS (PDB 7K3Y).

Calculations of zipper and LARKS propensity can be performed using our online web servers, which can be found at https://services.mbi.ucla.edu/zipperdb/ and https://srv.mbi.ucla.edu/LARKSdb/

# Field-specific reporting

Please select the one below that is the best fit for your research. If you are not sure, read the appropriate sections before making your selection.

☒ Life sciences    ☐ Behavioural & social sciences    ☐ Ecological, evolutionary & environmental sciences

For a reference copy of the document with all sections, see nature.com/documents/nr-reporting-summary-flat.pdf

# Life sciences study design

All studies must disclose on these points even when the disclosure is negative.

| | |
|---|---|
| Sample size | All ThT assays were performed with three experimental replicates, and are generally considered sufficient for plate-reader based experiments to determine a standard deviation. This sample size has also been used in previous studies by our group (Cao et al. Cryo-EM structure and inhibitor design of human IAPP (amylin) fibrils. Nat. Struct. Mol. Biol (2020); Cryo-EM structure of hIAPP fibrils seeded by patient-extracted fibrils reveal new polymorphs and conserved fibril cores) as well as other groups (Wördehoff et al. α-Synuclein Aggregation Monitored by Thioflavin T Fluorescence Assay. Bio Protoc. (2018)). |
| Data exclusions | For the Zipper Score calculations in Fig. 1, all sequences containing prolines were excluded from data analysis. ZipperDB is unable to calculate an energetic score for amyloid structures containing prolines due to their disruptive effects on beta-sheet stacking |
| Replication | All attempts at replication were successful. For ThT assays and Western blots, experiments were performed with three independent experimental replicates. For EM images, at least 3 images were taken per condition |
| Randomization | No randomization was necessary for this study because investigators were comparing quantitative data under well controlled conditions. No human or animal subjects were used. For all ThT assays, samples were loaded to a single 96-well plate, treated equally, and the results were recorded using an automatic plate reader, leading to objective results that do not require randomization. |
| Blinding | Investigators were not blinded because conditions were well controlled and quantitatively measured/calculated in an unbiased manner, and not dependent on subjective interpretation or analysis. |

# Reporting for specific materials, systems and methods

We require information from authors about some types of materials, experimental systems and methods used in many studies. Here, indicate whether each material, system or method listed is relevant to your study. If you are not sure if a list item applies to your research, read the appropriate section before selecting a response.

## Materials & experimental systems

| n/a | Involved in the study |
|---|---|
| ☐ | ☒ Antibodies |
| ☒ | ☐ Eukaryotic cell lines |
| ☒ | ☐ Palaeontology and archaeology |
| ☒ | ☐ Animals and other organisms |
| ☒ | ☐ Human research participants |
| ☒ | ☐ Clinical data |
| ☒ | ☐ Dual use research of concern |

## Methods

| n/a | Involved in the study |
|---|---|
| ☒ | ☐ ChIP-seq |
| ☒ | ☐ Flow cytometry |
| ☒ | ☐ MRI-based neuroimaging |

## Antibodies

| | |
|---|---|
| Antibodies used | i) Anti-Cytokeratin 8 mouse monoclonal antibody (C51); Cat # sc-8020, Lot # E1617, Santa Cruz Biotechnology. ii) β-Actin (C4) mouse monoclonal antibody . Cat # sc-47778. Lot # J1119. Santa Cruz Biotechnology. iii) Horseradish peroxidase (HRP)-conjugated goat anti-mouse IgG. Cat # ab205719. Lot # GR3405228-3 |
| Validation | i) Anti-Cytokeratin 8 mouse monoclonal antibody (C51); species reactivity: mouse, rat, human. Validation: Kim et al. JNK/SAPK mediates doxorubicin-induced differentiation and apoptosis in MCF-7 breast cancer cells. Breat Cancer Res. Treat. (2003). Application: Western blotting, immunoprecipitation, immunohistochemistry, solid phase ELISA. ii) β-Actin (C4) mouse monoclonal antibody; species reactivity: mouse, rat, human, avian, bovine, canine, porcine, rabbit, Dictyostelium discoideum and Physarum polycephalum. Validation: Zou et al. MAP4K4 induces early blood-brain barrier damage in a murine subarachnoid hemorrhage model. Neural. Regen. Res. (2021). Application: Western blotting, immunoprecipitation, immunohistochemistry, solid phase ELISA. |

