## [Peer Review File · Nature Structural & Molecular Biology]

Peer Review Information

Journal: Nature Structural and Molecular Biology

Manuscript Title: Identifying amyloid-related diseases by mapping mutations in low-complexity protein domains to pathologies

Corresponding author name(s): Professor David Eisenberg

Reviewer Comments & Decisions:

Decision Letter, initial version:
--

6th Aug 2021

Dear David,

Thank you again for submitting your manuscript "Identifying amyloid-related diseases by mapping mutations in low-complexity protein domains to pathologies". I apologize for the delay in responding, which resulted from the difficulty in obtaining suitable referee reports. Nevertheless, we now have comments (below) from the 3 reviewers who evaluated your paper. In light of those reports, we remain interested in your study and would like to see your response to the comments of the referees, in the form of a revised manuscript.

You will see that all reviewers are positive about the quality of the data and reviewers #2 and #3 are particularly supportive of publication of a revised manuscript. To this end, both reviewers have useful suggestions to improve the presentation and discussion of the findings, and to strengthen the experimental support for the claims made, such as adding data on the aggregation capacity of wildtype and mutant KRT8. Reviewer #1 is more critical and feels that recently published work compromises the novelty of the reported findings. Editorially, we disagree as we believe that the proposed mechanism will be of interest to our readership because of the potential relevance in disease, despite some conceptual overlap. Nevertheless, we would like to ask you to discuss this concern in the revised manuscript. Reviewer #3 also has some reservations about the support for the role of ethanol or liver tissue extracts from ASH patients in KRT8 aggregation and its inhibition using small molecules that should be addressed to their satisfaction.

Please be sure to respond to all concerns of the referees in full in a point-by-point response and highlight all changes in the revised manuscript text file. If you have comments that are intended for editors only, please include those in a separate cover letter.

We expect to see your revised manuscript within 8 weeks. If you cannot send it within this time, please contact us to discuss an extension; we would still consider your revision, provided that no similar work has been accepted for publication at NSMB or published elsewhere.

As you already know, we put great emphasis on ensuring that the methods and statistics reported in our papers are correct and accurate. **As such, I would like to stress that we will need you to submit both the Reporting Summary and the Editorial Policy Checklist with your revised paper.** Please follow the links below to download these files:

Reporting Summary:

Editorial Policy Checklist: <https://www.nature.com/documents/nr-editorial-policy-checklist.pdf>

Please note that the forms are dynamic 'smart pdfs' and must therefore be downloaded and completed in Adobe Reader.

When submitting the revised version of your manuscript, please pay close attention to our [href="https://www.nature.com/nature-research/editorial-policies/image-integrity">Digital Image Integrity Guidelines.](https://www.nature.com/nature-research/editorial-policies/image-integrity)

SOURCE DATA: we urge authors to provide, in tabular form, the data underlying the graphical representations used in figures. This is to further increase transparency in data reporting, as detailed in

this editorial (<http://www.nature.com/nsmb/journal/v22/n10/full/nsmb.3110.html>). Spreadsheets can be submitted in excel format. Only one (1) file per figure is permitted; thus, for multi-paneled figures, the source data for each panel should be clearly labeled in the Excel file; alternately the data can be provided as multiple, clearly labeled sheets in an Excel file. When submitting files, the title field should indicate which figure the source data pertains to. We encourage our authors to provide source data at the revision stage, so that they are part of the peer-review process.

Data availability: this journal strongly supports public availability of data. All data used in accepted papers should be available via a public data repository, or alternatively, as Supplementary Information. If data can only be shared on request, please explain why in your Data Availability Statement, and also in the correspondence with your editor. Please note that for some data types, deposition in a public repository is mandatory - more information on our data deposition policies and available repositories can be found below:

<https://www.nature.com/nature-research/editorial-policies/reporting-standards#availability-of-data>

We require deposition of coordinates (and, in the case of crystal structures, structure factors) into the Protein Data Bank with the designation of immediate release upon publication (HPUB). Electron microscopy-derived density maps and coordinate data must be deposited in EMDB and released upon publication. To avoid delays in publication, dataset accession numbers must be supplied with the final accepted manuscript and appropriate release dates must be indicated at the galley proof stage.

[REDACTED]

Note: This URL links to your confidential home page and associated information about

manuscripts you may have submitted, or that you are reviewing for us. If you wish to forward this email to co-authors, please delete the link to your homepage.

Kind regards,
Florian

Florian Ullrich, Ph.D.
Associate Editor
Nature Structural & Molecular Biology
ORCID 0000-0002-1153-2040

Referee expertise:

Referee #1: protein aggregation, X-ray crystallography

Referee #2: amyloids

Referee #3: protein aggregation, computational biology

Reviewers' Comments:

Reviewer #1:

Remarks to the Author:

Summary

The authors previously reported many predicted LARKS (low-complexity amyloid-like kinked segments) as responsible structural motifs for reversibility of functional amyloid-like fibrils of low-complexity domains in human proteome (Hughes et al., Science 2018). In the current manuscript, the authors identified disease-related mutations within the LARKS using three mutational databases and computationally screened them for the most likely mutations that increase the propensity to form a pleated beta-sheet typifying steric zipper conformation of pathogenic amyloid proteins as opposed to a kinked one observed in the LARKS. Among the resulting mutations, the authors picked up two disease mutations (G62C and G55A) in the head domain of Keratin-8 (KRT8) intermediate filament protein, which is known to aggregate pathologically in the form of Mallory Denk bodies (MDBs) in liver disease. They determined amyloid fibril structures of KRT8 peptides containing the disease mutations. The

authors found that the mutations cause transition in backbone from an extended conformation to a pleated one, which increases interactions between neighboring subunits and thereby stabilizes the mutant amyloid structure. The authors also examined effects of alcohol on KRT8 aggregation since the MDBs are most frequently observed in alcoholic steatohepatitis (ASH) and alcoholic cirrhosis. They show that the KRT8 peptides bind to ethanol or isopropanol in the crystal structure. In addition, aggregation of the wild-type KRT8 head domain was promoted in the presence of ethanol. The authors showed that tissue extract from ASH patients enhance aggregation of recombinant KRT8 head domain apparently by a seeding effect. They tested some compounds that are known to reduce amyloid aggregation, and found that antibiotic demeclocycline HCl seems to reduce KRT8 aggregation.

Originality

McKnight group at UT Southwestern Medical Center has already reported that the head domains of Desmin and Neurofilament light (NFL) intermediate filament proteins and the tail domain of fly Tm1 intermediate filament protein can form amyloid-like fibrils, and that disease-causing mutations in Desmin and NFL facilitate formation of amyloid-like fibrils of the head domains (Zhou et al., PNAS 2021; Sysoev et al., PNAS 2020). Thus, the originality of the research in this manuscript is not high.

Data and methodology

Computational work and structural determination work seem to be of good quality. Regarding all the thioflavin-T assays, there is no error bar and it is not clear that how many replications were carried out for each sample.

Conclusions: robustness, validity, reliability

Effects of alcohol:

KRT8 peptides were crystallized with ethanol or 2-propanol as a precipitant. Thus, it is not surprising that these alcohol molecules were trapped in a crystal lattice (Fig 4ab). The ThT assays with ethanol (Fig 4c) used the KRT8 head domain (90 aa), which is different from the peptide segments used for crystallization. Therefore, it is not clear whether the trapped alcohol molecules are the basis for facilitated aggregation observed in the ThT assays.

Seeding effects with liver tissue extracts:

Two of the three ASH samples showed initial signal increase, but soon later, the signal decreased and reached down to the same level of the HCC sample and "KRT8 alone". It is confusing and seems to be risky to simply interpret that ASH extracts facilitate aggregation of KRT8 head domain.

Small molecule inhibition:

Did the "KRT8 alone" contain the same amount of DMSO with other samples (Fig 4f)? The signal differences between "KRT8 alone" and the other samples were relatively small in comparison to the noise level, thus it is important to replicate the samples and obtain error bars.

Reviewer #2:

Remarks to the Author:

In this report, Murray et al expand their work from a 2018 Science paper to find low complexity sequences in the human proteome that are candidates for amyloid-like assembly using a computational threading scheme. Here they extend their approach to disease-associated protein variants and identify variant keratin sequences that are predicted to take on a more robust assembly state (“zipper”) than their wildtype counterpart, based on secondary structural considerations. Atomic level structural determination of mutant and wildtype peptides was consistent with this idea. The effects of alcohol and small molecule inhibitors were studied, as was a seeding experiment with patient liver extracts.

Strengths:

- *This paper is a creative and productive use of a computational screen to find potential amyloid-altering protein variants associated with disease.
- *The effect of alcohol upon KRT8 1-90 aggregation is strong and unequivocal. Its relatedness to Mallory Denk bodies should be an interesting future investigation.
- *The seeding experiments were also satisfying and suggestive of an in vivo correlate of the findings from the paper.
- *The data showing small molecule inhibition of aggregates was interesting and convincing.

Weaknesses:

- *The use of threading to find LARKS was published and keratins fell out of that analysis. So this part is not new.
- *The use of hnRNPA1 served as a stopgap for the inability to solve one of the wildtype peptide’s structure. Suboptimal, but understandable.
- *The aggregation capacity of full length keratin 8, for the wildtype and pathogenic variant, would make the report more compelling.

Other comments

-the parts of figure 3 are mislabeled in the text. The disorder probability portion of the figure (3a in the figure itself) is not referred to in the text and the subsequent parts of the figure are shifted, there was also an absence of reference to 3d in the text.

-The fluorescent aggregates seen in figure 3e appear amorphous in contrast to the liquid droplets in the cited work on keratins (refs 24, 25). The authors should discuss this difference.

-Underlining is not necessary for the last paragraph

-delete 'of' in the sentence: "Additionally, evidence of these extended conformations of can be found in the full-length fibril structures of FUS and TDP43 (Supp. Fig. 10)."

In sum: This report is interesting, timely, and resourceful. A possible clinical connection is intriguing. The data largely support the conclusions and a potential wider application of the computational screen is promising.

Reviewer #3:

Remarks to the Author:

Eisenberg and co-workers made predictions, based on computational modeling, for mutations in a low complexity region of the intermediate filament protein Keratin-8 (KRT8) that would drive amyloid aggregation. This approach is based on their hypothesis that mutations in reversibly aggregating proteins such as KRT8 may promote pathogenic irreversible aggregation by driving a transition from a kinked to pleated beta-sheet conformation which favors steric zipper formation. They confirm their hypothesis for three KRT8 mutants: G55A, G62C and Y54H. Moreover, they show that ethanol promotes KRT8 aggregation, which is not the case for alpha-synuclein. This is of relevance for alcoholic liver disease where aggregated KRT8 is found in Mallory-Denk bodies, and that KRT8 aggregation can be seeded with liver extract of such patients, which allows to classify this disease as an amyloid-related condition. Finally, the authors identified 2 compounds with possible therapeutic effects: EGCG and the antibiotic demeclocycline.

This manuscript is of high relevance as it revealed several novel findings, especially that i) mutations in low complexity regions can be detrimental to favoring amyloid aggregation by encouraging a pleated beta-sheet conformation, ii) alcoholic liver disease can be classified as an amyloid-related condition, iii) the nutraceutical EGCG may counteract the KRT8 aggregation-encouraging effect of ethanol.

Moreover, both the quantity of work performed and the quality of the study are at highest levels. The manuscript is clearly and concisely written.

I have only minor comments for the authors:

1) l. 185: I cannot fully agree to the statement "the three mutants aggregated much more quickly and extensively compared to wild-type", as for the WT there is no lag time visible (Fig. 3b). Thus, it seems that the WT is already aggregated at the start of the experiment, i.e. it seems to aggregate the fastest. I agree that it appears to aggregate less.

2) l. 189: If the distance between the strands is smaller than usual (4.3-4.6 Å vs. 4.8 Å), wouldn't this imply a stronger hydrogen bond between the strands of the beta-sheet? And does the identity of the side chains may play a role here? For example, might the presence of several Gly residues encourage such a smaller distance. It would be good if the authors could comment on this.

3) l. 224: It is interesting that ethanol promotes the aggregation of KRT8, whereas it has no or the opposite effect on alpha-Syn. Can the authors provide a physicochemical explanation for this finding?

4) l. 272: There is a word missing in "Additionally, evidence of these extended conformations of can be found..." Please correct.

Author Rebuttal to Initial comments

We thank the reviewers for their help in improving and clarifying our paper. We address each of their points, as follows:

Reviewer #1:**Originality**

McKnight group at UT Southwestern Medical Center has already reported that the head domains of Desmin and Neurofilament light (NFL) intermediate filament proteins and the tail domain of fly Tm1 intermediate filament protein can form amyloid-like fibrils, and that disease-causing mutations in Desmin and NFL facilitate formation of amyloid-like fibrils of the head domains (Zhou et al., PNAS 2021; Sysoev et al., PNAS 2020). Thus, the originality of the research in this manuscript is not high.

Response: We thank Reviewer #1 for highlighting the important recent work by the McKnight and Murray groups, and we now reference these papers in the Discussion. Whereas both the previous work by McKnight/Murray and our current study focus on the low-complexity domains of specific intermediate filament proteins and associated mutations, the source of novelty of our study is its generalized methodology. In our approach the proteins and mutations of interest are identified by an unbiased structure-based algorithm to screen for LARKS motifs that contain mutations which make them more prone to form amyloid steric zippers. This approach can be applied to identify novel amyloid pathologies throughout the proteome. We demonstrate the power of our method by identification of the intermediate filament protein KRT8 for further characterization based on its high predictive score in our computational model. We are encouraged that the prior work referenced by Reviewer 1, which analyzes the amyloid nature of the head/tail domains of Desmin, NFL, and Tm1, and associated aggregation-inducing mutations, corroborate our findings for the head domain of KRT8. We anticipate that our method will enable other scientists to readily identify other amyloid conditions and their associated pathogenic proteins.

Data and methodology

Computational work and structural determination work seem to be of good quality. Regarding all the thioflavin-T assays, there is no error bar and it is not clear that how many replications were carried out for each sample.

Response: Error bars have been added to all ThT assays reported, and updated in the following figures: Fig. 3b, Fig. 4c-g, Supp. Fig. 2. Experimental replicate numbers have also been reported in the corresponding figure captions.

Conclusions: robustness, validity, reliability

Effects of alcohol: KRT8 peptides were crystallized with ethanol or 2-propanol as a precipitant. Thus, it is not surprising that these alcohol molecules were trapped in a crystal lattice (Fig 4ab).

The ThT assays with ethanol (Fig 4c) used the KRT8 head domain (90 aa), which is different from the peptide segments used for crystallization. Therefore, it is not clear whether the trapped alcohol molecules are the basis for facilitated aggregation observed in the ThT assays.

Response: The co-crystallization of the KRT8 peptide segments with ethanol and 2-propanol were initially coincidental findings. Hundreds of different crystallization conditions were tested for both peptide segments and it happened that the conditions that led to crystallization of both segments contained alcohols. This is what led us to further test the effect of alcohol in aggregation of the KRT8 head domain. We agree with Reviewer 1 that the exact binding the alcohols to the peptide segments seen in the crystal structures may not precisely represent the mode of interaction alcohol has with the head domain/full-length protein. Instead, these structures provide initial insight into the KRT8-alcohol relationship. We find the crystallographic results, in addition to the ThT assays of KRT8/tau/aSyn with ethanol, compelling evidence that alcohol has some type of affinity for KRT8 over other amyloid proteins. This is especially notable given the strong correlation of MDBs (i.e. KRT8 aggregates) to alcoholic liver disease. Further investigation will be needed to determine to exact nature mechanism by which alcohol potentiates KRT8 aggregation, which is beyond the scope of this current study. We have updated the manuscript to reflect this, (see Results, under sub-heading “The effects of ethanol and the seeded aggregation of KRT8 by ASH liver tissue”).

Seeding effects with liver tissue extracts:

Two of the three ASH samples showed initial signal increase, but soon later, the signal decreased and reached down to the same level of the HCC sample and “KRT8 alone”. It is confusing and seems to be risky to simply interpret that ASH extracts facilitate aggregation of KRT8 head domain.

Response: It is not unusual for Thioflavin T fluorescence to rise sharply on amyloid seeding and then fall over time. This frequently observed behavior is thought to arise from fibril maturation. Thioflavin T fluoresces as it is bound in the crevice of amyloid protofilaments. In some cases, as protofilaments couple and twist to form mature fibrils, some of the Thioflavin T is excluded, lowering total fluorescence.

Small molecule inhibition:

Did the “KRT8 alone” contain the same amount of DMSO with other samples (Fig 4f)? The signal differences between “KRT8 alone” and the other samples were relatively small in comparison to the noise level, thus it is important to replicate the samples and obtain error bars.

Response: Yes, the “KRT8 alone” sample contained the same amount of DMSO vehicle as the maximum demeclocycline containing sample (1 μ M), to a final DMSO concentration of 0.1%. We have updated the figure label to just “KRT8”. The caption for Fig. 4g (formerly Fig. 4f) and Methods section has been updated to clarify this. Error bars are now included in Fig. 4g, and caption updated to include number of experimental replicates.

Reviewer #2:

Strengths:

- *This paper is a creative and productive use of a computational screen to find potential amyloid-altering protein variants associated with disease.
- *The effect of alcohol upon KRT8 1-90 aggregation is strong and unequivocal. Its relatedness to Mallory Denk bodies should be an interesting future investigation.
- *The seeding experiments were also satisfying and suggestive of an in vivo correlate of the findings from the paper.
- *The data showing small molecule inhibition of aggregates was interesting and convincing.

Response: We thank Reviewer #2 for encouraging comments.

Weaknesses:

- *The use of threading to find LARKS was published and keratins fell out of that analysis. So this part is not new.

Response: Our prior work by Hughes and colleagues (ref #6) did initially identify several keratin proteins as containing LARKS, but no further analysis/structure determination was performed. The novelty of our current study lies in its, a) evaluation of the role pathogenic mutations play in LARKS-containing proteins, b) confirmation of the presence of LARKS-like motifs in Keratin, c) identification that KRT8 aggregates in an amyloid-like manner and, d) draws new correlation of KRT8 aggregation to liver disease, not previously considered an amyloid disease.

- *The use of hnRNPA1 served as a stopgap for the inability to solve one of the wildtype peptide’s structure. Suboptimal, but understandable.

Response: True. Many attempts were made to determine the wildtype crystal structure. High resolution diffraction data were collected, but we encountered difficulty with phasing. While not optimal, we believe the sequences of the hnRNPA1 segment reported by Gui et al. (ref #24) and the wildtype KRT8 segment are similar enough that meaningful conclusions can be drawn.

*The aggregation capacity of full length keratin 8, for the wildtype and pathogenic variant, would make the report more compelling.

Response: Our report addresses a topic of intense current interest: the interaction of low-complexity protein domains in disease. For that reason, we have focused on the low-complexity domain of keratin 8, avoiding complications that could be introduced by including additional domains. In making this choice we are following the lead of McKnight and other investigators in this field. We agree with the implication of Reviewer #2 that the one of the next steps towards understanding of the in vivo processes is to work with the full protein. The complication likely to be encountered is protein solubility, which is achieved in cells by the action of numerous other macromolecules and solvent molecules.

Other comments

-the parts of figure 3 are mislabeled in the text. The disorder probability portion of the figure (3a in the figure itself) is not referred to in the text and the subsequent parts of the figure are shifted, there was also an absence of reference to 3d in the text.

Response: We thank Reviewer #2 for identifying this error. The disorder probability predictions for Fig. 3a are now mentioned in the Results section of the main text. Reference to Fig. 3d is now made in the main text. The subsequent references to Fig. 3 in the main text have also been corrected.

-The fluorescent aggregates seen in figure 3e appear amorphous in contrast to the liquid droplets in the cited work on keratins (refs 24, 25). The authors should discuss this difference.

Response: We have added a sentence, "The fluorescent aggregates appear more amorphous than some other condensates, perhaps due to experimental variability, as the imaged sample was in a larger bulk volume of a plate well, not on a glass slide (see Methods), or perhaps the amorphous appearance is resultant from transition to a more solid/fibrillar aggregate state, not pure liquid phase separation, in our conditions." (see Results, "Characterization of the aggregation-promoting effects of disease-associated KRT8 mutations")

-Underlining is not necessary for the last paragraph

Response: The underlining has been removed from the mentioned section in the manuscript. (Discussion, paragraph 8)

-delete 'of' in the sentence: "Additionally, evidence of these extended conformations of can be found in the full-length fibril structures of FUS and TDP43 (Supp. Fig. 10)."

Response: Thank you. This error has been corrected in the manuscript (Discussion, paragraph 3)

In sum: This report is interesting, timely, and resourceful. A possible clinical connection is intriguing. The data largely support the conclusions and a potential wider application of the computational screen is promising.

Response: We thank Reviewer #2 for this comment, and agree that a potential clinical correlation to liver disease found in this study merits further investigation, and further computational screening has potential to identify additional previously unsuspected amyloid pathologies

Reviewer #3:

This manuscript is of high relevance as it revealed several novel findings, especially that i) mutations in low complexity regions can be detrimental to favoring amyloid aggregation by encouraging a pleated beta-sheet conformation, ii) alcoholic liver disease can be classified as an amyloid-related condition, iii) the nutraceutical EGCG may counteract the KRT8 aggregation-encouraging effect of ethanol. Moreover, both the quantity of work performed and the quality of the study are at highest levels. The manuscript is clearly and concisely written.

We thank Reviewer #3 for these comments.

1) l. 185: I cannot fully agree to the statement "the three mutants aggregated much more quickly and extensively compared to wild-type", as for the WT there is no lag time visible (Fig. 3b). Thus, it seems that the WT is already aggregated at the start of the experiment, i.e. it seems to aggregate the fastest. I agree that it appears to aggregate less.

Response: Qualitatively, the numbers of WT fibrils present on the electron micrographs compared to the three mutants were far fewer. The aggregation curve of the WT KRT8₁₋₉₀ (Fig. 3b, black curve) is partially masked by the large magnitude of the mutant protein curves (Fig. 3b, red/blue/green). Thus, we also include a zoomed-in version of the WT KRT8 aggregation data as Supp. Fig 2. This shows that aggregation of the sample occurs after ~24h and at a slower rate than the mutant proteins.

2) I. 189: If the distance between the strands is smaller than usual (4.3-4.6 Å vs. 4.8 Å), wouldn't this imply a stronger hydrogen bond between the strands of the beta-sheet? And does the identity of the side chains may play a role here? For example, might the presence of several Gly residues encourage such a smaller distance. It would be good if the authors could comment on this.

Response: This is a good question, and the answer gets into a subtle effect, probably not appropriate for discussion in this paper. The separation of the strands is the same as in other beta sheets, ~ 4.8 Å and the hydrogen bonds have the same lengths. But in the LARKS structures, there is essentially a screw axis, leading to extinction of the 4.8 Å reflection. Another common separation of 4.6 Å shows up in the powder diffraction. To avoid the confusion produced by the present text, we have removed two sentences, and now simply state that the spacings of 10 and 4.3-4.6 Å are typical of LARKS and low-complexity protein aggregates.

3) I. 224: It is interesting that ethanol promotes the aggregation of KRT8, whereas it has no or the opposite effect on alpha-Syn. Can the authors provide a physicochemical explanation for this finding?

Response: Reviewer #3 asks a tantalizing question, but prediction and interpretation of protein solubilities is a challenging undertaking. The solubility of a substance is given by the free energy difference of the dissolved and solid forms. Prior to experiments, we would not have guessed that the most soluble amino acid in water at neutral pH is proline (presumably because of the high free energy of crystalline proline), whereas the least soluble amino acid is tyrosine (presumably because it forms a stable multiply H-bonded (low free energy) crystal). To explain the relative aggregation potentials of KRT8 and alpha-synuclein in the presence of ethanol would be a study in itself. Nothing better illustrates the challenge of computing protein solubilities than the method adopted by crystallographers to determine the conditions for protein crystallization—namely massive empirical screening of conditions, routinely hundreds or even thousands of conditions. Though the method is expensive, it works, whereas computation does not.

4) I. 272: There is a word missing in “Additionally, evidence of these extended conformations of can be found...” Please correct.

Response: Thank you. This error has been corrected in the manuscript. The sentence now reads: “Additionally, evidence of these extended conformations can be found in the full-length fibril structures of FUS and TDP43 (Supp. Fig. 11)”

Decision Letter, first revision:

15th Dec 2021

Dear David,

Thank you again for submitting your revised manuscript "Identifying amyloid-related diseases by mapping mutations in low-complexity protein domains to pathologies" (NSMB-A45000A). It has now been seen by the original referees and their comments are below. The reviewers find that the paper has improved in revision, and therefore we'll be happy in principle to publish it in Nature Structural & Molecular Biology, pending minor revisions to satisfy the referees' final requests and to comply with our editorial and formatting guidelines.

Kind regards,
Florian

Florian Ullrich, Ph.D.
Associate Editor
Nature Structural & Molecular Biology
ORCID 0000-0002-1153-2040

Reviewer #1 (Remarks to the Author):

All the points that I raised concern for were properly addressed by the authors. The manuscript is now ready for publication.

I found some minor errors in supplementary figures.

- 1) Supp Fig 1. There is no color indicator (like a one in Fig1C).
- 2) Supp Fig 2 figure legend. "1-90" in KRT81-90 should be subscript.
- 3) Supp Fig 7. Although the end of the figure legend said "Error bars represent +_ SD", I don't see any error bars in the figure.

Reviewer #2 (Remarks to the Author):

The only comment I can add is that both I (Rev 2) and reviewer #1 mentioned the reduced novelty of this MS based on prior publication on the topic from both these authors and that of the McKnight/Murray groups. The introductory paragraph could be clearer about how this report differs from prior work - something like the text response to the first 'weakness' bullet for Reviewer 2.

Reviewer #3 (Remarks to the Author):

The authors carefully addressed the comments of all reviewers. I have no further queries.

Author Rebuttal, first revision:

Responses to Reviewers

Reviewer #1

All the points that I raised concern for were properly addressed by the authors. The manuscript is now ready for publication.

I found some minor errors in supplementary figures.

- 1) Supp Fig 1. There is no color indicator (like a one in Fig1C).
- 2) Supp Fig 2 figure legend. "1-90" in KRT81-90 should be subscript.
- 3) Supp Fig 7. Although the end of the figure legend said "Error bars represent +_ SD", I don't see any error bars in the figure.

Response:

We thank the Reviewer for their constructive comments regarding the publication, and address their minor comments below:

- 1) Supp. Fig. 1 (now Extended Data Fig. 1) has been updated to include a color indicator for the heat map.
- 2) Supp. Fig. 2 (now Extended Data Fig. 2), the text “1-90” is now in subscript.
- 3) Supp. Fig. 7 (now Extended Data Fig. 7), now contains error bars.

Reviewer #2

The only comment I can add is that both I (Rev 2) and reviewer #1 mentioned the reduced novelty of this MS based on prior publication on the topic from both these authors and that of the McKnight/Murray groups. The introductory paragraph could be clearer about how this report differs from prior work - something like the text response to the first 'weakness' bullet for Reviewer 2.

Response:

We thank the Reviewer for their comments and agree that the findings by the McKnight and Murray groups are important and relevant to this current study. We now reference their work (Sysoev et al. 2020; Zhou et al. 2021) in the Introduction section (Line 77), as follows:

“Recent reports have demonstrated that low complexity domains in several intermediate filament proteins undergo amyloid assembly (Sysoev et al. 2020; Zhou et al. 2021)”

Additionally, we mention their work in the Results (Line 123) to better clarify their previous findings with respect to ours:

“While other intermediate filament proteins have been demonstrated to form amyloids, KRT8 specifically has not previously been characterized as an amyloid-forming protein.”

Final Decision Letter:

8th Apr 2022

Dear David,

We are now happy to accept your revised paper "Identifying amyloid-related diseases by mapping mutations in low-complexity protein domains to pathologies" for publication as a Article in Nature Structural & Molecular Biology.

To assist our authors in disseminating their research to the broader community, our SharedIt initiative provides all co-authors with the ability to generate a unique shareable link that will allow anyone (with

or without a subscription) to read the published article. Recipients of the link with a subscription will also be able to download and print the PDF.

As soon as your article is published, you can generate your shareable link by entering the DOI of your article here: http://authors.springernature.com/share.

Corresponding authors will also receive an automated email with the shareable link

Note the policy of the journal on data deposition:

<http://www.nature.com/authors/policies/availability.html>.

Your paper will be published online soon after we receive proof corrections and will appear in print in the next available issue. You can find out your date of online publication by contacting the production team shortly after sending your proof corrections. Content is published online weekly on Mondays and Thursdays, and the embargo is set at 16:00 London time (GMT)/11:00 am US Eastern time (EST) on the day of publication. Now is the time to inform your Public Relations or Press Office about your paper, as they might be interested in promoting its publication. This will allow them time to prepare an accurate and satisfactory press release. Include your manuscript tracking number (NSMB-A45000B) and our journal name, which they will need when they contact our press office.

About one week before your paper is published online, we shall be distributing a press release to news organizations worldwide, which may very well include details of your work. We are happy for your institution or funding agency to prepare its own press release, but it must mention the embargo date and Nature Structural & Molecular Biology. If you or your Press Office have any enquiries in the meantime, please contact press@nature.com.

If you have not already done so, we strongly recommend that you upload the step-by-step protocols used in this manuscript to the Protocol Exchange. Protocol Exchange is an open online resource that allows researchers to share their detailed experimental know-how. All uploaded protocols are made freely available, assigned DOIs for ease of citation and fully searchable through nature.com. Protocols can be linked to any publications in which they are used and will be linked to from your article. You can also establish a dedicated page to collect all your lab Protocols. By uploading your Protocols to Protocol Exchange, you are enabling researchers to more readily reproduce or adapt the methodology you use, as well as increasing the visibility of your protocols and papers. Upload your Protocols at www.nature.com/protocolexchange/. Further information can be found at

www.nature.com/protocolexchange/about.

Please note that *Nature Structural & Molecular Biology* is a Transformative Journal (TJ). Authors may publish their research with us through the traditional subscription access route or make their paper immediately open access through payment of an article-processing charge (APC). Authors will not be required to make a final decision about access to their article until it has been accepted. [Find out more about Transformative Journals](https://www.springernature.com/gp/open-research/transformative-journals)

Authors may need to take specific actions to achieve [compliance with funder and institutional open access mandates](https://www.springernature.com/gp/open-research/funding/policy-compliance-faqs). If your research is supported by a funder that requires immediate open access (e.g. according to [Plan S principles](https://www.springernature.com/gp/open-research/plan-s-compliance)) then you should select the gold OA route, and we will direct you to the compliant route where possible. For authors selecting the subscription publication route, the journal's standard licensing terms will need to be accepted, including [self-archiving policies](https://www.springernature.com/gp/open-research/policies/journal-policies). Those licensing terms will supersede any other terms that the author or any third party may assert apply to any version of the manuscript.

Kind regards,
Florian

Florian Ullrich, Ph.D.
Associate Editor
Nature Structural & Molecular Biology
ORCID 0000-0002-1153-2040